# Energy and the Macrodynamics of Agrarian Societies

Georgios Karakatsanis [1,2,*] and Nikos Mamassis [1]

1 Department of Water Resources and Environmental Engineering, School of Civil Engineering, National Technical University of Athens (NTUA), 9 Heroon Polytechneiou St., 15870 Zografou, Greece; nikos@hydro.ntua.gr
2 Department of Research, EVOTROPIA Ecological Finance Architectures Private Company (P.C.), 190 Syngrou Avenue, 17671 Kallithea, Greece
* Correspondence: georgios@hydro.ntua.gr; Tel.: +30-69-4555-2243

**Abstract:** For the present work, we utilized Leslie White's anthropological theory of *cultural evolutionism* as a theoretical benchmark for econometrically assessing the *macrodynamics* of *energy use* in *agrarian societies* that constituted the human civilization's second *energy paradigm* between 12,000 BC and 1800 AC. As White's theory views a society's ability to harness and control energy from its environment as the *primary function of culture*, we may classify the evolution of human civilizations in three phases according to their energy paradigm, defined as the *dominant pattern of energy harvesting from nature*. In this context, we may distinguish three energy paradigms so far: *hunting–gathering*, *agriculture*, and *fossil fuels*. Agriculture, as humanity's energy paradigm for ~14,000 years, essentially comprises a secondary form of solar energy that is biochemically transformed by photosynthetic life (plants and land). Based on this property, we model agrarian societies with similar principles to natural *ecosystems*. Just like natural ecosystems, agrarian societies receive abundant solar energy input but also have limited land ability to transform and store them biochemically. As in natural ecosystems, this constraint is depicted by the *carrying capacity* emerging biophysically from the *limiting factor*. Hence, the historical dynamics of agrarian societies are essentially reduced to their struggle to *maximize energy use by maximizing the area and productivity of fertile land –in the role of a solar energy transformation hub– mitigating their limiting factor*. Such an evolutionary forcing introduced *technical upgrades*, like the leverage of domesticated livestock power as a *multiplier* of the caloric value harvested by *arable* and *grazing* land combined. According to the above, we tested the econometric performance of four selected dynamic maps used extensively in ecology to reproduce humanity's energy harvesting macrodynamics between 10,000 BC and 1800 AC: (a) the *logistic* map, (b) the *logistic growth* map, (c) a lower limiting case of the Hassel map that yields the *Ricker* map, and (d) a higher limiting case of the Hassel map that yields the *Beverton–Holt* map. Following our results, we discuss thoroughly our framework's major elaborations on social hierarchy and competition as mechanisms for allocating available energy in society, as well as the related future research and econometric modeling challenges.

**Keywords:** cultural evolutionism; macrodynamics; agrarian society; energy paradigm; ecosystems; logistic map; logistic growth map; Hassel map; Ricker map; Beverton–Holt map; limiting factor

## 1. Introduction

The theory of *cultural evolutionism* was developed by anthropologist Leslie White [1] in an attempt to establish an analytical framework able to reduce the rise and fall of human civilizations. In the context of cultural evolutionism, the *primary function of culture* is *available energy* for the generation of thermomechanical work. A human civilization's determinant to generate thermomechanical work is the available *energy technology*; defined as *all the artificial structures built by humans for transforming and/or diverting natural energy fluxes towards their collective formations*. As White specifically noted [1], "*Social systems are determined by technical systems*". His views echoed a quite earlier theory by anthropologist

Lewis Henry Morgan [2], who considered technological progress as a prerequisite of social evolution, focusing on Ancient Greece and Rome, where historical records were relatively sufficient. Although Morgan did not propose a concise reductionist model with energy at its core, he postulated a set of causal relationships between the establishment of *family property* and *technological advancement* for the formation of a *bottom-up* (grassroots) social hierarchy process.

In addition, White considered the rapid collapse of the Western Roman Empire as the best case study for proving his theory on the link between available energy and social complexity. Indeed, due to the existence of analytical records on city growth, agricultural output, and productivity—that being the energy currency of agrarian societies—for the first century AC [3], he was able to extrapolate a relationship between the growth of the empire and the minimum amount of energy (in terms of agricultural output) required to sustain it. In accordance with the above works, anthropologist Joseph Tainter expanded White's conclusions, developing a generalized energy economic theory of *marginal productivities* in order to explain the rapid collapse of the Western Roman Empire, as well as that of more than twenty civilizations of various size and internal organization sophistication—like the Mayans and Chacoans—for a period extending over 2000 years [4]. Although lacking a solid mathematical framework, the added value of Tainter's work can be found in the fact that he contributed to a consistent *universal* theory on the bond between *energy* and *structural complexity* at the socio-historical level, verifying numerous works integrating natural and social systems under the *second law of thermodynamics* [5–9].

### 1.1. Energy Paradigm, Structural Change, and Social Organization

Although viewing the historical course of human societies with similar physical or statistical mechanical principles to (open) thermodynamic systems provides an empirically accurate analytical and explanatory tool irrespective of space and time, the crucial distinction for the study of large-scale and long-term social structures concerns the identification of their *dominant pattern of energy harvesting from nature*. In this context, we may establish the definition of the *Energy Paradigm* [5,10], which classifies the historical course of human societies *primarily* according to the *reference natural resource* used to achieve and maintain the level of organization as a *necessary condition* and *secondarily* according to the potential of the social covenant to sustainably distribute the available energy harvested from the environment to the population of the society as a *sufficient condition*. Tainter [4] clearly demonstrated that the collapse of the Western Roman Empire can be mainly attributed to the inefficiency of the state to achieve a distribution that would motivate its subjects to maintain the empire's state of complexity at the time. When these inefficiencies are combined to external stressors as well, so that even the flow of energy from the natural source is disrupted (e.g., in the case of Rome due to invasions of Germanic tribes), societies become unable to maintain their complexity and start to decompose structurally. The above input preserves the general context of *open thermodynamic systems* and further enriches it in terms of studying the internal structures of social systems with concepts of "phase change", which are consistent with the framework used to examine the relation between energy and structural complexity in physical systems.

According to the above, modeling the energy *macrodynamics* of human social systems primarily concerns their classification according to the dominant pattern of energy harvesting from nature in order to achieve a both broad and solid sense of homogeneity. As presented in Figure 1, we may distinguish three *Energy Paradigms* across the recorded and established anthropological history, accepting the appearance of Neanderthal species around 350 kaBC as our starting point. The first energy paradigm is identified in *Hunter–Gatherer* societies that were based on human muscle energy. These societies were based on the short-term satisfaction of energy needs with low storage and accumulation capacity (one week or less), as well as high mobility and low numbers ranging between 30 and 100 people [11,12]. The energy currency of hunter–gatherer societies was secondary solar energy in the form of the stored biochemical energy of gathered and collected plants, as

well as herbivore and carnivore wildlife, positioned at higher levels of the trophic pyramid (above plants), presenting higher-nutritional-value hunting opportunities (such as protein inputs).

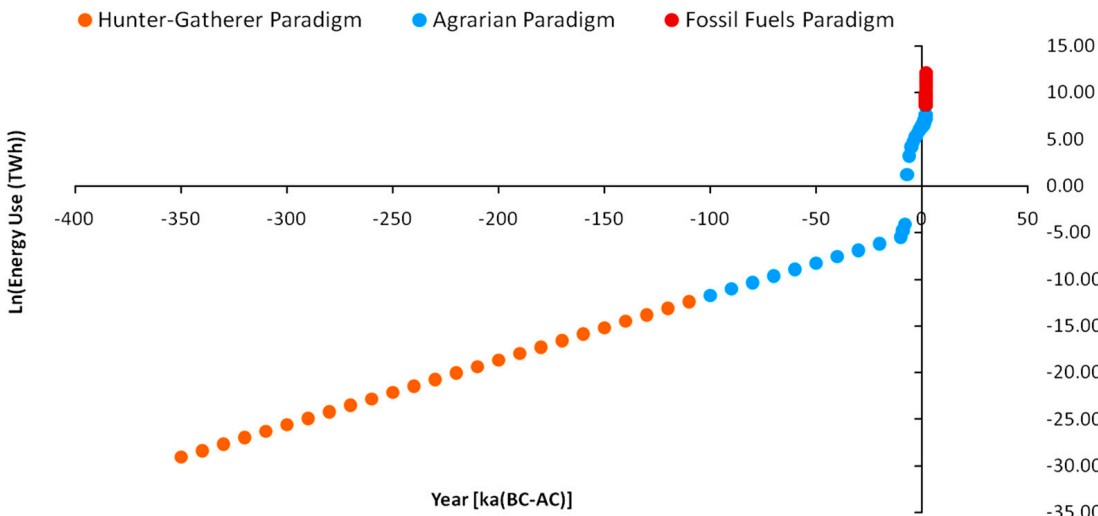

**Figure 1.** Natural logarithm (Ln) of energy use (TWh) across the three identified *Energy Paradigms* in human history: (**a**) *Hunter–Gatherer Paradigm* (350 kaBC–12 kaBC; orange); (**b**) *Agrarian Paradigm* (12 kaBC–1800 AC; blue); and (**c**) *Fossil-fueled Paradigm* (1800 AC–to date; red). While the availability of energy increases exponentially, the periods of transition decrease exponentially.

The natural successors of hunter–gatherers were *Agrarian* societies, which are also the object of this study. The first agricultural societies exhibited a structural shift in both dimensions of the energy paradigm around 12,000 BC, as most anthropological records demonstrate [13–16]. Regarding the shift in the dominant pattern of harvesting energy from nature, the shift involved the transformation of natural land, which sustained primitive societies in a rather random biogeophysical pattern, into *arable* and *grazing* land, where crops could be planted and harvested *at will* and also human labor could be utilized to maximize the yield of biochemically stored solar energy. While hunter–gatherers were in a marginal state of energy availability and constantly at high risk of decomposition, agrarian societies achieved to establish an output level above the minimum required for exact reproduction and maintenance (steady-state *subsistence*) [17–20]. The main result of this new paradigm was the creation of gross food *surpluses*, the equivalent of *capital accumulation* today (the flagship of *economic growth*). This shift inevitably affected the social relationships for the distribution of these surpluses. During the agrarian energy paradigm, the concepts of private property [21] and social hierarchy [22] emerged and were established for the first time in history. These core concepts led to the generation of many others, such as the allocation of work, salaries, social classes, bureaucracy, output monitoring, and taxation, as derivatives encapsulating and protecting the core idea of *private wealth* [1,2]. Examining the agrarian paradigm, we can also observe the first large-scale transformations of the natural environment and related anthropogenic impacts [23].

The third and most recent energy paradigm concerns *Fossil-fueled Industrial* societies, which evidence humanity's latest shift since the beginning of the *Industrial Revolution*. In principle, the shift in resources consisted in the utilization of fossil remnants of dead organisms that have existed since the appearance of photosynthetic life (3.8 GaBC). Fossil fuels were extracted as products of very long geological processes of extreme temperature and pressure conditions and were further distilled, diversified, and concentrated in hydrocarbon deposits of very high energy densities [24]. Typically, fossil fuels are also a secondary form of solar energy, embodied in organisms via the food chain, and, at death,

molecularly decomposing before being re-structured into the various categories and hydro-carbon quality classes (solid, liquid, and gas) via energy inputs from physicochemical and geological processes that purify the compounds and upgrade their thermal content [25–27]. Typically, the latest paradigm should include nuclear fuels that are also geologically extracted. However, despite the fact that the use of nuclear energy began in the last quarter of the ongoing energy paradigm, its global primary energy use share is still only around 4% [28], with fossil fuels remaining dominant in the global energy mix, although nuclear energy will probably comprise humanity's fourth energy paradigm in the future.

In short, the energy history of human civilizations may be described as the successive transition from one energy paradigm to the next. Figure 1 separates the anthropological eras according to their adopted energy paradigm. For this reconstruction, we utilized raw aggregate data for the *fossil-fueled* paradigm [28], while for the *agrarian* paradigm we combined raw data regarding land use [29] from the HYDE 3.2 database reconstruction [30]. These data were combined with current estimations on required land inputs for the production of various foods with caloric value of 1000 kcal each [31] based on the methods presented in [32]. Due to a lack of more accurate estimations, these methods were assumed to be applicable to the agrarian paradigm as well. For the *hunter–gatherer* paradigm, the lack of data was severe, however of secondary importance; hence, we assumed an exponential growth of energy use since 350 kaBC so that the first reconstructed value would be yielded at 10,000 BC. These datasets and reconstruction methods were further used to structure the macrodynamic models that will be thoroughly presented in subsequent sections. In addition, points after year 1800 AC in Figure 1 are denser due to the increased availability and frequency of recorded data beyond this year. An important conclusion that can also be drawn from Figure 1 is that, while across the energy paradigm transitions, the energy availability increases exponentially, the periods of transition decrease exponentially. More analytically, we may see this feature in Figure 2 below, with combined data from [6,28]:

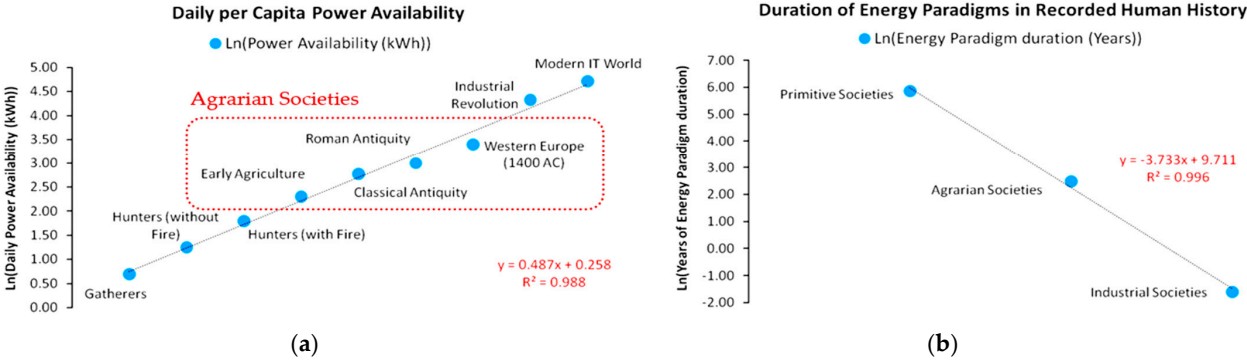

(**a**)                                                                (**b**)

**Figure 2.** Average per capita power availabilities in each energy paradigm and duration of energy paradigms in natural logarithm (Ln) scale: (**a**) Per capita power availabilities from hunter–gatherers to modern *Information Technology* (IT) intensive industrial societies. The marked window encapsulating the period from early agriculture to pre-industrial societies highlights the agrarian energy paradigm; (**b**) duration of each energy paradigm. In the 12,000 years of agrarian societies, the average person had more than 10 times more available energy than his hunter–gatherer predecessor (almost 338 ka), while the average person in a modern industrial society in the last 220 years has access more than 100 times more available power than the average person in an agrarian society.

According to Figure 2, with the fossil-fueled paradigm, in just 220 years, humanity has used more energy than all the energy used by the hunter–gatherer and agrarian paradigm combined (across a period of 350 ka). The catalytic element was the invention and commercial deployment of the *internal combustion engine*, for which fossil fuels comprised an ideal *complementary* economic good. For instance, although the various uses of petroleum (e.g., medicine, military) were already established across the agrarian paradigm, the technologies capable of maximizing its productive power were absent. Furthermore, the social

elements that originally appeared in agrarian societies–such as private property and social hierarchy–remained and even intensified to a global scale due to the productivity of energy in internal combustion engines, which, combined to mechanical capital, started massively claiming the productivity share of manual labor [24]. A major consequence was also the establishment of the large-scale deployment of *credit*, as these energy surpluses not only allowed the detachment of fiat currencies from agricultural output but also directed massive amounts to future ventures on new technological advancements with highly uncertain yield [33–35]. Finally, environmental impacts also intensified and globalized. The scarcity of deposits or carrying capacities was realized [36,37], practically establishing the field of natural resource economics. In turn, even scientific discussions on the establishment of a new geological era based on the energetic anthropogenic footprint on interlocked planetary biogeochemical cycles have taken place ever since [38]. An additional structural shift was the transformation of industrial agriculture from *net energy producer* to *net energy user* via its heavy dependence on petrochemical fertilizers [39,40]. Although the above depictions are quite useful for understanding the sequence of energy paradigms, the thorough analysis of this unprecedented energy availability for industrial societies in terms of both resource use and internal social structure is out of the scope of the current work and, as a result, has been put aside for future work.

### 1.2. Energy and the Ecodynamics of Civilizations

From the works of Morgan, White, and Tainter, we may postulate that, in the growth and collapse of human civilizations, irrespective of their energy paradigm, almost every feature of *ecosystem dynamics* can be identified as a socio-physical analog [25–27]. In particular, the cycles of growth, stability, and collapse observed in past civilizations can be modeled via similar principles that apply to complex thermodynamic systems that continue to grow beyond the limits of their energy budget at subsistence state (i.e., the state of exact sustenance of the system's structures and exact replenishment of their wear-out). The alternative path is a successful transition to a new paradigm with higher resources' abundance. Figure 3 depicts two indicative energy paradigm transition processes.

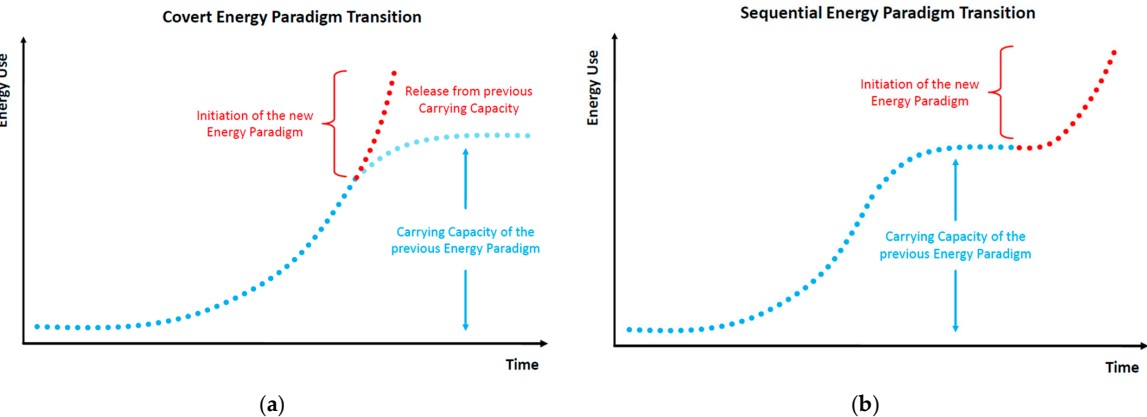

(**a**)　　　　　　　　　　　　　　　　　　　　　　　　　　(**b**)

**Figure 3.** Schematic depiction of two standard processes of energy paradigm transitions: (**a**) *Covert* and (**b**) *Sequential*. In both cases, the first energy paradigm relies on a renewable energy source (e.g., agrarian societies fueled by solar energy), so upon reaching a maximum capacity (e.g., of land use), it stabilizes due to the continuous external (by the Sun) replenishment of metabolized energy. In the *covert* transition, the initiation of the new energy paradigm occurs before the old meets its carrying capacity, giving the overall impression of an exponential growth pattern. This is the pattern observed in our civilization [28]. The *sequential* transition occurs after the civilization has met its current energy paradigm's carrying capacity, which reveals its upper limit and true pattern of logistic growth. Numerous other cases with carrying capacity fluctuations or collapses are also relevant as special cases that have empirically been observed, although for simplicity are not presented.

In this context, we may introduce two core concepts—(a) *Capacity* and (b) *Metabolism*—as analogs to natural ecosystems [41]. These concepts relate a civilization's energy paradigm to its internal structures' efficiency to optimally allocate physical work [5–10], transcend the current paradigm's biophysical limitations, and enter a new paradigm that is based on resources of higher abundance. Energy use growth within an energy paradigm follows similar principles to ecosystem populations and can be modeled by continuous or discrete time logistic growth functions. All logistic growth functions have the universal mathematical feature of being distinguishable in three phases, irrespective of the differences in the duration of each phase according to the chosen model (e.g., Verhulst, Gompertz). Similarly, as shown in Figure 3, an energy paradigm can be separated into three periods: *formation*, *acceleration* and *saturation*. The *formation* period concerns the initial conditions for the self-organization of the norms regulating the energy paradigm and is essentially a product of long-term social ferments. The *acceleration* period concerns the exponential segment of the curve, reflecting its rapid adoption and eventual dominance as energy harvesting pattern. The *saturation* period concerns the growth curve part following the inflection point, with diminishing growth rate due to diminishing unutilized carrying capacity, eventually stabilizing the paradigm at an upper limit.

Typically, energy paradigms are initiated by a *structural change* event which usually manifests intensively near or along the saturation period of the previous energy paradigm; although not spontaneously. Structural change is usually the result of long-term social ferments, constituting the factors that trigger energy transitions throughout human history. As according to White [1], energy is the primary function of culture, setting the primary limits of a civilization's growth, its institutions comprise a complete set of algorithms regulating the internal flow of available energy harvested from the environment, i.e., its *social metabolism*. Indicative cases of such institutions are population control legislation, R&D, the monetary system, political regimes, the market competition structure, trade networks, etc. In Figure 3, the facet of social metabolism is depicted by the curve's growth rate. Higher social metabolism generates a steeper growth path, meaning that the carrying capacity is met faster. However, it is important to acknowledge R&D, a universal evolutionary factor across all energy paradigm transitions. Indeed, we may identify early forms of R&D that led the transition from hunter–gatherers to agrarian societies. Such a case was the allocation of work between sexes, with males mainly dealing with high-risk hunting activities and females with the organization of the camp's space, classification of seeds, and empirical optimization of food sources, which eventually led to the empirical application of agricultural practices.

Furthermore, in Figure 3, the horizontal axis represents the time range in which an energy paradigm is adopted, while the vertical axis represents the energy use scale. The energy use scale and, in turn, carrying capacity depend on the civilization's efficiency $h$, with $h \in (0, 1)$, in metabolizing every unit of available primary energy into useful physical work while minimizing thermodynamic losses at life-cycle. The energy paradigm duration depends on the intensity of energy use growth at each level of energy efficiency $h$. It is well understood that, with thermodynamic losses being inevitable due to the universal validity of the *second law of thermodynamics* [5–10], maximizing the fraction of the theoretical potential of an energy paradigm depends on a very sensitive dynamic equilibrium between the energy use growth rate and the adoption rate of increasingly energy efficient technical inputs. Hence, although, from a purely ecological aspect, intuitively, the system's maximization of energy and rapid growth might be considered as its primary target, the maximization of useful work across the energy paradigm's life-cycle may require a more conservative and gradual growth pattern. There are practically infinite different combinations between energy use growth and technological improvement patterns. The change in the parameters of the energy use growth pattern changes the growth structure as well, extending or diminishing each one of the energy paradigm's periods. However, a crucial element that is missing from continuous-time logistic growth models but exists in the discrete-time models is that excessive energy use growth rates may overshoot carrying

capacity, risking a civilization's sustainability, growth and further evolution, which by no means should be considered secure, as the collapse of the Western Roman Empire has shown [4]. Contrarily, a civilization's sustainability and evolution highly depends on its ability to metabolize available primary energy into useful work and direct it via sophisticated social algorithms to its population. This mathematical property regarding the *intrinsic energy use growth rate* has substantial economic meaning as well.

## 2. Materials and Methods

In this section, we thoroughly describe the adopted methodological framework, data collection sources, and data transformation methods used for structured modeling. Specifically, this section consists of three main dimensions of our methodology, separated in respective paragraphs: (a) A theoretical examination of the relation between energy and social complexity as identifiable elements in agrarian societies after 8000 BC through the large-scale domestication of animals, (b) an empirical examination of selected dimensions of growth in agrarian societies according to the reconstructed HYDE 3.2 data [30], and (c) the mathematical framework of discrete-time growth maps that have been used to reproduce energy use growth for the time period 10,000 BC–1800 AC.

### 2.1. Energy and Socio-Ecological Complexity

Irrespective of an energy paradigm's special attributes, the analysis so far has substantiated a universal property of social systems: with a source of constant energy flow and sufficient energy surpluses, social systems follow similar organization principles to open thermodynamic systems, increasing their *structural complexity* [10]. The upper limit of social complexity will be theoretically discussed in the subsequent sections. Regarding agrarian societies, it is important to highlight that *land* comprises the equivalent of internal combustion engines in industrial societies, as, via plant biomass, it transforms incoming primary solar energy to storable secondary biochemical energy. At a social level, arable and grazing land are combined to leverage the power of livestock and maximize available energy. This structure forms a *socio-ecological energy pyramid* similar to Lindeman's classic trophic pyramid [42], as shown in Figure 4.

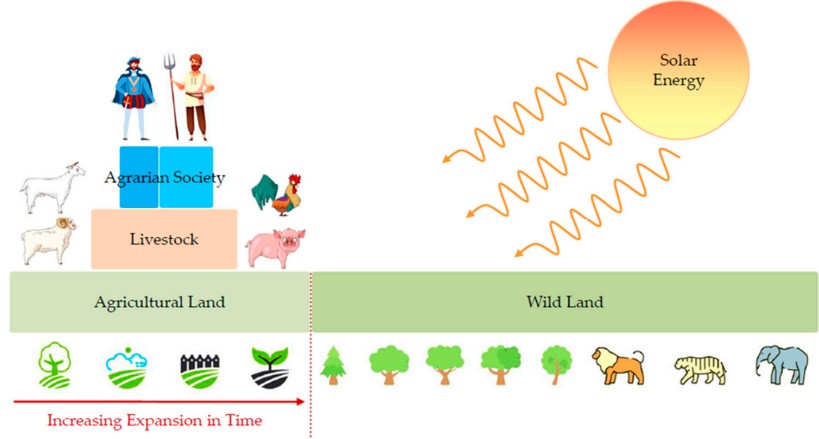

**Figure 4.** Schematic depiction of the socio-ecological trophic pyramid of energy flow and distribution in an agrarian society. At the physical level, the agrarian energy paradigm consists of the increasing transformation of *wild* land to *arable* and *grazing* land (accounting for *total agricultural land)* for harnessing secondary solar energy in the form of storable biochemical energy of crops that will be further channeled and distributed to society by the social covenant between the classes.

Figure 4 provides insight into how a hierarchical agrarian system of available energy distribution optimization is formed from its biophysical foundations. Starting from the socio-ecological pyramid's base, a tipping point for the transition from hunter–gatherers to agricultural economies was the classification of the various seeds as a primary form of

science and research via long-term observation and trial and error. The first step of such a transition was the transformation of wildland and its increasing displacement. Across this land transformation, the HYDE 3.2 data [30] reveal another important feature. While, for a period of 2000 years, from 10,000 BC to 8000 BC, we can observe positive *built-up area* values for settlements and *cropland area*, it is only after 8000 BC that we can observe the first positive *grazing area* values. Combining data with the literature [43,44], it is fair to assume that, between 10,000 and 8000 BC, the agricultural output depended almost exclusively on human manual labor. This does not exclude the primary form of the domestication of animals; however, research suggests that it was rather local and small-scale. Although, in relation to hunter–gatherers, these societies enjoyed higher available energy levels, they remained at a level of *subsistence*, with slow growth rates.

The second phase of agrarian social systems concerned their internal re-structuring and formation of stronger spatiotemporal social hierarchies in relation to the subsistence state. The large-scale domestication of animals signified a major diversification of production methods, skyrocketed productivity, and leveraged the available energy potential. In turn, this ignited a process of higher complexity in social relations. In particular, from 1000 BC to 1800 AC, human labor was estimated to provide a capacity of just 75 W. Contrarily, the capacity of domesticated horses increased for the same period from just 296 W to 1155.5 W via breeding optimization, signifying an increase of over 290% [33,34]. The time required to till an area of 1 ha exclusively by human labor reached 400 h, almost 16.6 days, while with the use of oxen pair accounted for just 65 h, i.e., 2.7 days, signifying a productivity increase of 514% [34]. Following such impressive energy efficiency increases, domesticated animals and livestock became an integral part of the agrarian socio-ecological pyramid as second-level solar energy transformers with *net energy yield*.

Furthermore, as shown in Figure 5, estimates from early agricultural New Guinea [44] show a life-cycle energy investment of kcal 224,520.2/ha distributed in more than 12 different inputs. The surplus energy yield across harvesting is estimated to be more than 16 times, leading to an *Energy Return on Energy Invested* (EROEI) of 16:1 and a *Net Energy Gain* (NEG) of 15. In more detail, more than 53% of the invested energy of ~225,000 kcal concerns only two forms of work—*Planting and Weeding* (32.1%) and *Cartage* (21.3%)—the latter of which is practically transportation. With a 16-fold surplus of 89,810,080 kcal, the total caloric output in this 400 ha area would be able to sustain a population of 123 individuals with daily needs of 2000 kcal each. This population is quite rational to assume as, in 1963–64, the estimated sustained population in the area was just 200 individuals with their pig flocks. Typically, as a pig's typical daily caloric needs are ~5440 kcal [45], we can fairly assume that the human population size in this early agrarian society was ~40–50% of the 1962–1963 records as the residual caloric budget sustained the pigs' population.

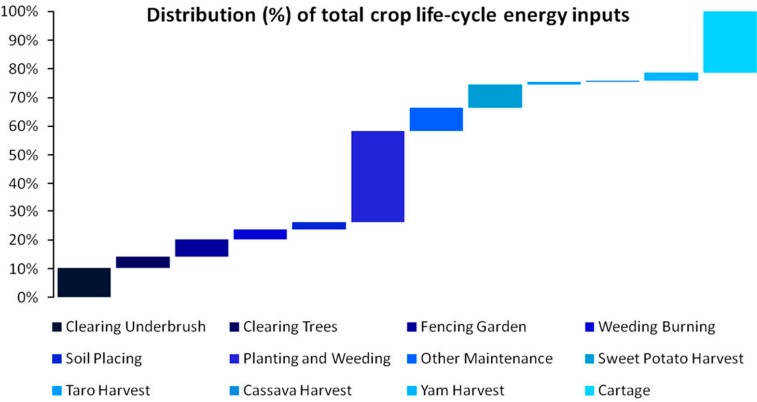

**Figure 5.** Distribution of human labor life-cycle energy input fractions per *ha* across a cropland in primitive New Guinea for 12 different crop stages. Sequence of works is presented across columns in the nomenclature (S1: Clearing Underbrush; S2: Clearing Trees; S3: Fencing Garden; etc.).

The above case introduces a crucial concept for the analysis of agrarian societies, the *Energy Equivalent Population* (EEP). The EEP is defined as the *maximum sustainable population of specific daily caloric needs level*. The EEP is typically a specific *energy budget individual* (either human or animal) that, as a concept, provides us with flexibility to chart the numerous combinations between different species and produce the optimal ones. Specifically, for a $n$ number of species that comprise the elements of an agrarian society (humans and domesticated animals), with a specific caloric need $\varepsilon$ of each individual of each species $i$ and a maximum caloric budget $E$ (capped) at time step $t$, the maximum $EEP_i$ (if all of the energy budget was dedicated to sustain only one species $i$) in an agrarian system that is either in subsistence or has established an operating energy budget to produce a specific amount of surpluses is:

$$\overline{EEP}_{it} = \frac{\overline{E_t}}{\varepsilon_i}, \quad E, \varepsilon \in R^+ \tag{1}$$

Equation (1) depicts the maximum number of sustainable *EEP* individuals for every species $i$ at every time step $t$. For the distribution of the total energy budget $E$ (capped) in more than one species, in order to estimate each species share of the total caloric budget, we may write:

$$p_{it} = \left( \frac{\varepsilon_i \cdot N_{it}}{E_t} \right), \quad p \in (0,1], N \in N^+ \tag{2}$$

Equation (2) transforms EEPs to total energy budget shares by multiplying the specific caloric needs of each species by its physical population $N$ (number) at time $t$ and dividing it by the total energy budget $E$ (capped). This normalization allows for the application of *Information Entropy* metrics within a probabilistic framework. Summing budget shares of all species $n$ gives the total energy budget.

$$\sum_{i=1}^{n} p_{it} = 1, \quad n \in N^+ \tag{3}$$

Based on the above, we adopt the formulation of *Renyi Entropy* [46] as a *generalization of Shannon Entropy*. From an *Information Theory* perspective, we may use *Renyi Entropy* to depict the total energy budget's composition via the probability density of each species in the agrarian society's species' mix. Hence, the Renyi Entropy for an agrarian society consisting of discrete elements (here distinct species) is:

$$H_q = \frac{1}{(1-q)} \cdot ln \sum_{i=1}^{n} p_i^q, \quad q \in R^+ - [1] \tag{4}$$

For the asymptotic convergence given for a parameter value $q = 1$, the Renyi Entropy becomes the typical *Shannon Entropy* for discrete systems [46]:

$$\lim_{q \to 1} H_q = H(X) = -\sum_{i=1}^{n} p_i \cdot ln p_i \tag{5}$$

According to Equations (4) and (5), through using *Information Entropy*, we can statistically interpret the concentrations of species in the total energy budget as well as other concentration concepts. Here, we can model the species forming the total energy budget mix with normalized EEPs that reduce them into comparable energy units and apply the *Shannon Entropy* formula in a straightforward way. Hence, if there are $n$ elements (species) that form a total energy budget mix ($E = \sum \varepsilon \cdot N$), the *entropy maximization* of the mix occurs for the *exact same probability* (equiprobability) to meet any of the energy budget's species. For this special case—in which all elements have the same probability $p_i$, in order to be

found by a process of random selection—with $p_1 = p_2 = \ldots = p_i = 1/E$, as an equivalent to $\varepsilon_1 \cdot N_1 = \varepsilon_2 \cdot N_2 = \ldots = \varepsilon_i \cdot N_i$, Equation (5) becomes:

$$H(E)_{MAX} = -ln\left(\frac{1}{E}\right) = -ln p_i \qquad (6)$$

Equations (1)–(6) constitute the mathematical framework for assessing the complexity of agrarian socio-ecological pyramids (Figure 4). Complexity levels reflect human diet, economic diversity and technology. For instance, while in the case of the early agrarian New Guinea the caloric structure was simple—consisting of only 4 crops and pigs—from ancestry to pre-industrial times, the pyramid became much more complex, consisting of both a variety of crops and animals [3,47,48], with the latter used for a variety of scopes, such as labor, war, food, and vesture. Each pyramid structure option has cost and benefits; simpler pyramids reflect less advanced societies and more equal caloric distributions, while complex pyramids reflect higher sophistication (e.g., targeted breeding for protein intake), intense social hierarchy, and infrastructures of higher required EROEI to sustain long-term social complexity.

### 2.2. Energy and Growth in Agrarian Societies

In this part, we use selected data from the HYDE 3.2 database [30,49] to extrapolate relationships concerning population and land use growths for the period 10,000 BC–1800 AC. According to the available data, Figure 6 presents the correlations between population, cropland, grazing land, and built-up land growths for the examined period, as well as the correlation between the global population and total agricultural land—as the sum of cropland and grazing land. As presented in the correlation combinations below, Figure 6a–d suggests strong positive correlations between the growth of the global human population and the growth in all 3 land use types, verifying the theoretical depiction of Figure 4 on the expansion of agricultural land over wildland.

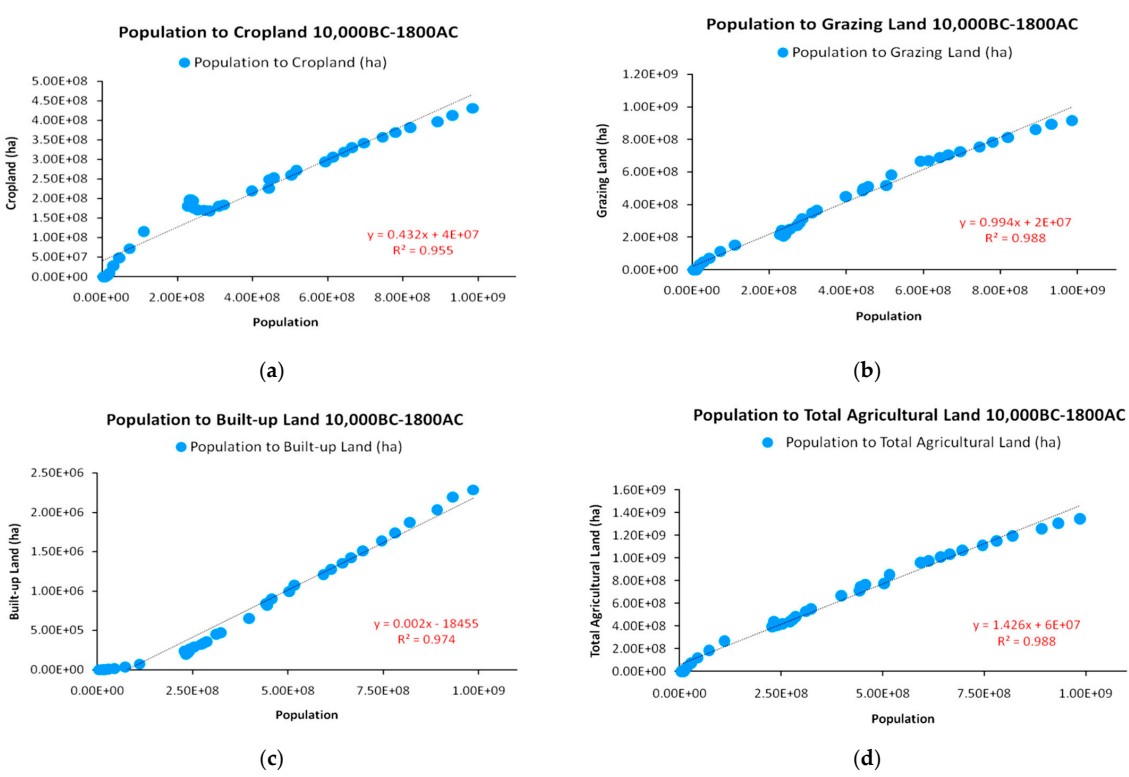

**Figure 6.** Correlations for 10,000 BC–1800 AC between human population and various land uses (**a**) cropland; (**b**) grazing land; (**c**) built-up land; (**d**) total agricultural land (based on HYDE 3.2 data).

These correlations confirm the fundamental bond between energy availability and population growth, irrespective of energy paradigm, as also shown in Figure 2a. In addition, this is a fundamental hypothesis in dynamic population growth models, such as the logistic growth models presented in Figure 3 and developed in Appendix A for studying the reproduction of EEP growth dynamics. The core concept is that populations will continue to grow in the presence of residual (unutilized) carrying capacity or when carrying capacities increase via technological upgrades. The growth of caloric yield per unit land is such a technical upgrade.

As already mentioned, according to HYDE 3.2 data, it was not until 8000 BC that the large-scale domestication of animals took place, where the transformation of wild land to grazing land comprised the initial energy invested for sustaining livestock for improved caloric and protein yield. The correlation between the cropland and grazing land growth (shown in Figure 7) verify this fundamental assumption on the evolution of energy flow complexity in agrarian systems. In particular, we focus on the strong correlation between cropland and grazing land, as across the transition from subsistence and the dependence on human labor for output of crops, a major shift was the re-direction of a fraction of crop production to animals to sustain a necessary biomass of increased labor productivity, as presented in Section 2.1.

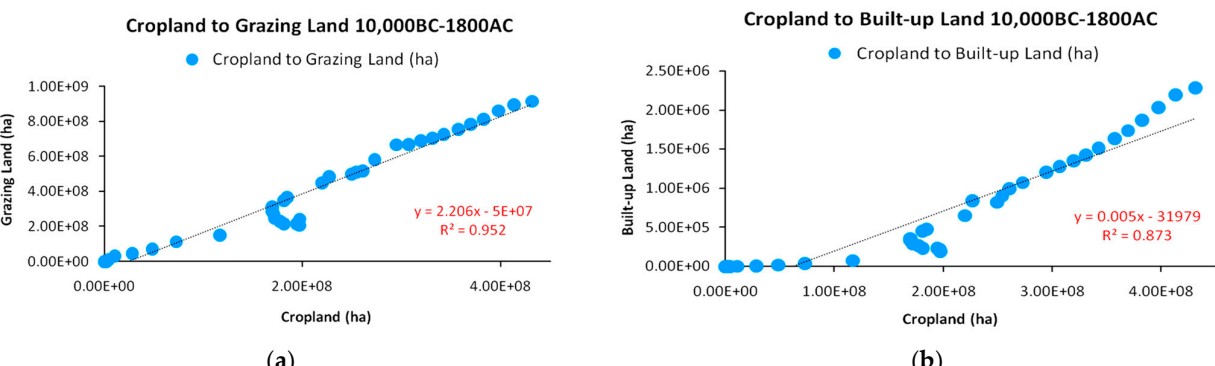

**Figure 7.** Correlations for 10,000 BC–1800 AC between (**a**) cropland to grazing land growth and (**b**) cropland to built-up land growth (based on HYDE 3.2 data).

Although, according to Figure 7, the lowest correlation in the HYDE 3.2 data proved to be between cropland and built-up land for settlements ($R^2 = 0.873$), the high correlations in Figure 8 provide a corrective and more explanatory picture. In early agrarian societies with small-scale animal domestication, settlements could be much smaller in size in order to fence a minimum arable area as security stock and more efficiently defend against raids. However, after 8000 BC, the average size of settlements became significantly larger [25,27,33–35] in order to provide infrastructure for the large populations of domesticated animals. Another aspect of large-scale animal domestication and increase in total agricultural land (with cropland and grazing land as its elements) in relation to built-up land were large infrastructure projects such as roads and aqueducts for the transfer of critical water resources for irrigation over long distances to sustain such a level of agricultural production and road networks for the safe transportation and trade of produced commodities. Although, in a strict sense, built-up land concerns settlements and cities, these infrastructures should be considered as inseparable parts of that land use category. As shown in Section 2.1, the use of domesticated animals provided an unprecedented power input and efficiency increase in terms of saved time, and it is highly doubtful that agrarian societies would have otherwise met that level of growth after ancestry.

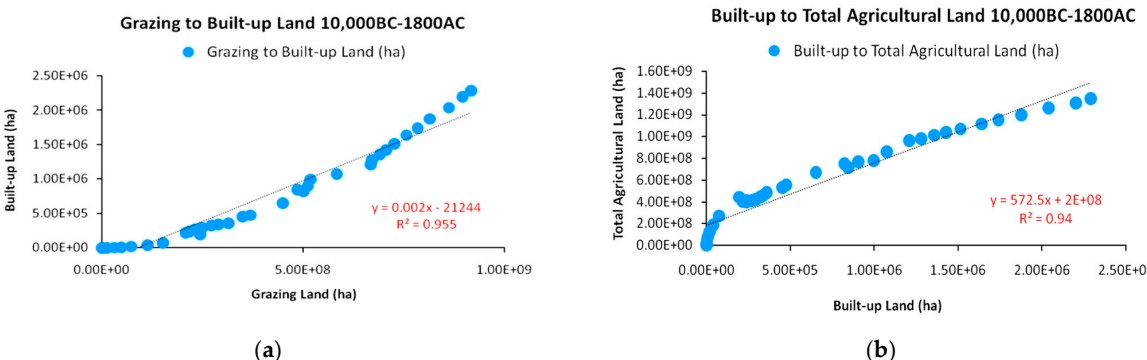

**Figure 8.** Correlations for 10,000 BC–1800 AC between (**a**) grazing to built-up land growth and (**b**) built-up to total agricultural land growth (based on HYDE 3.2 data).

*2.3. Agrarian Energy Paradigm Structure and Resource Distribution*

The correlations between various combinations of human population and land uses provide a strong indication of a correlation between energy availability and growth. An additional dimension examined in this part is the temporal dynamics between cropland and grazing land by the *Fisher–Pry* ratio on technological substitution [50]:

$$s_{iF-P} = \frac{s_i}{1 - s_i}, \quad s \in [0, 1), i \in N^+ \tag{7}$$

Equation (7) provides the standard Fisher–Pry relation on technological substitution in terms of the adoption shares $s$ of each technology $i$. The results derived from the HYDE 3.2 database on cropland and grazing land ratios and distributions are presented in Figure 9. In Figure 9a we show the cropland/total agricultural land ratio for the whole duration of the agrarian paradigm between 10,000 BC–1800 AC, along with the empirical and simulated (fitted) normal distributions. It is notable that 50% (=19) of total available observations (=38) show an empirical value of 0.35 (=35%) of cropland to total agricultural land ratio, while 85% (=32) of the observations suggest a range of the cropland/total agricultural land ratio between 0.35 and 0.5 (35–50%) with very high positive skewness (=1.6595). Remaining values concern low and high outliers. Moreover, we may observe from the graph that while, in 8000 BC, cropland completely dominates agricultural land (=100%), its share from 7000 BC starts diminishing to ~83%, and after 6000 BC (the point at which animal domestication and reproduction became a more common practice), the share of cropland until 1800 AC neither falls below 24% nor increases above 50%, with a mean value of 35%. In economic terms, the shares of cropland/total agricultural land and of cropland/grazing land—as partially competitive and partially complementary technologies—remain relatively constant across population and land use growth.

A respective view is presented in Figure 9b for the Fisher–Pry ratio between cropland and grazing land, although less intensively. Specifically, 39.4% (=15) of total available observations (=38) show an empirical value of 0.5 (=50%) ratio (exactly at the theoretical mean), while only 47.3% (=18) of observations are in the range of 0.35–0.5 (35–50%), with extremely high positive skewness (=5.3383). However, the skewness is heavily affected by the outlier value in 7000 BC, where cropland was 4.78 times higher than grazing land. A sample of values from 8000 BC provides a much lower skewness (=1.0288) and a more symmetric distribution.

Considering that cropland and grazing land are *partially complementary* and *partially competitive* technologies (one type of land cannot be completely substituted by the other, while a minimum part of cropland yield will be re-directed to domesticated animals that cannot rely exclusively on grazing), we may examine further evidence on the scaling of energy surpluses and the net energy gains via the large-scale domestication of animals. In Figure 10, we present the estimated daily caloric needs of various animal species and

human individuals. The values shown were composed by a combination of historical estimations and reconstructed data [6,15,25–27,33–35,43,51–53] with rational assumptions. Figure 10 represents the animals that were widely domesticated and used throughout the agrarian energy paradigm. In addition, we can observe a significant inequality of caloric needs between human individuals in different social classes. In particular, a lactating cow may be metabolizing up to 28,000 kcal to maintain its body mass and functions, while in a non-lactating cow, this value may increase up to 21,500 kcal. Typical labor or transportation horses have daily needs of 20,000 kcal. It is fair to assume that a war horse trained to carry heavy armor and engage in combat would require a significantly higher daily caloric intake. Moreover, a pig has daily caloric needs of ~5500 kcal, a shearing sheep 100 has needs of 2500 kcal, a large dog has needs of 800 kcal, a hen has needs of 300 kcal, and a cat has needs of just 200 kcal.

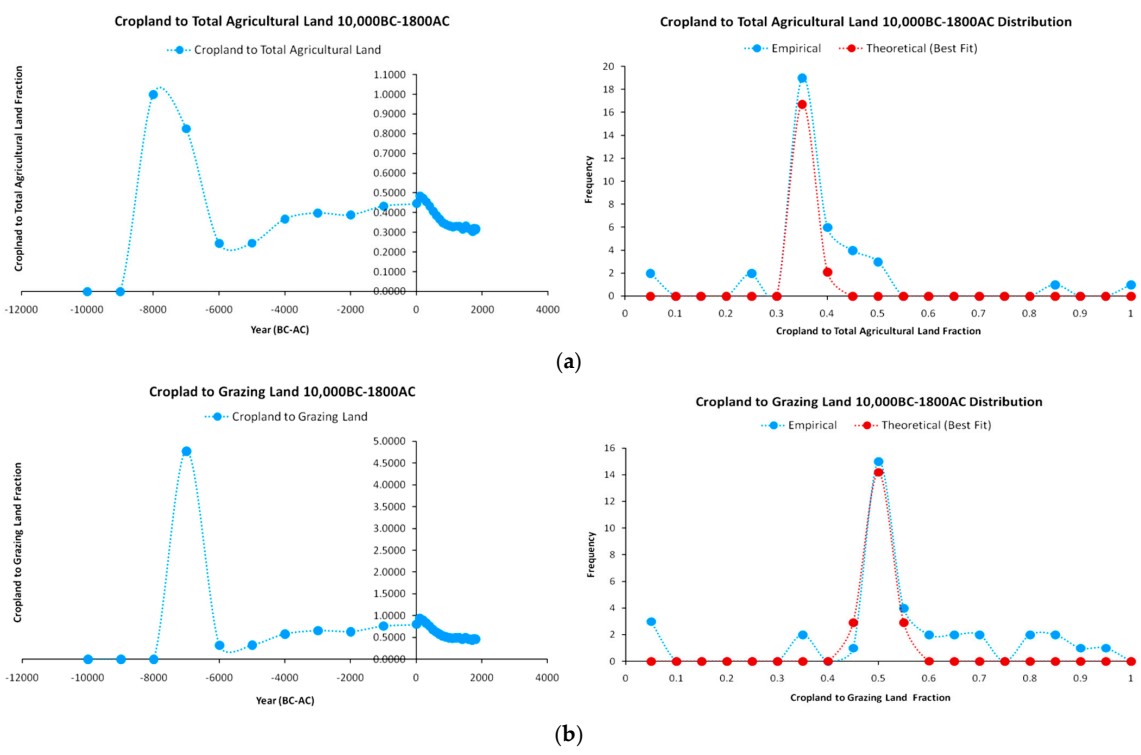

**Figure 9.** Fisher–Pry relations and statistical distributions for (**a**) cropland to total agricultural land and (**b**) cropland to grazing lands for 10,000 BC–1800 AC (based on HYDE 3.2 data).

A much more interesting insight was revealed for the daily caloric needs of human individuals at various stages of the agrarian energy paradigm and at different social and economic states. With the lack of internal combustion engines and modern technology, a typical farmer had to complete daily farm work solely via manual labor (even the steering of animals for tillage should be viewed as such). A reference farmer would have had a daily caloric need of 3000 kcal, a one-third over today's recommended average daily caloric intake of 2000 kcal, which is near the standard UN refugee camp ration of 1700 kcal. However, although farmers constituted the majority of human populations during the agrarian energy paradigm, significant variability across social classes may be identifiable. For instance, as a multi-discipline soldier, a Roman legionnaire would have required a daily caloric intake of 6000 kcal, and a medieval monk would have required a daily caloric intake of 5500 kcal, while a medieval aristocrat required 4500 kcal daily. Such differences suggest an intra-social caloric variability up to 100% higher than the standard. Due to the lack of specific data, it is difficult to estimate the intra-social shares of classes in human populations and estimate the respective EEPs following the framework of Equations (1)–(7) without resorting to rational assumptions and simulations.

**Typical Daily Energy Needs (kcal)**

**Figure 10.** Estimated daily caloric needs (kcal) of various animal species and human individuals in natural logarithm (Ln) scale in descending order at various periods, according to social and economic state. Estimations are based on current diets for animals and reconstructed data for humans.

In any case, the most interesting insight concerns social complexity in relation to the leveraging of domesticated animal power for net energy gains and the social covenants forming the stratification of classes, as presented in Figure 4. By combining the findings in Figures 9 and 10, along with data presented in Section 2.1, we argue that, from a socio-ecological perspective, via the domestication of animals, agrarian systems maximized amounts of secondary solar energy from crops to escape subsistence and grow and diversify their internal structure [54,55]—respectively to biological species in natural ecosystems [56]. In fact, these principles can be identified even in modern subsistence rural societies, which, in part, provide a temporal window into the past [54,57]. Specifically, at around 1000 BC (at a time of limited animal domestication), a horse provided ~3.95 times more power (296 W) than human labor (75 W), with its daily caloric intake being 7 times higher than the typical human diet of 3000 kcal (if assumed constant). Overall, a horse would require a net energy *opportunity cost* of ~3 humans, meaning that each horse would be chosen for breeding instead of 3 humans (usually slaves). By 1800 AC, a horse's power provision increased to 1155.5 W, i.e., by 290% [33,34], so that its productivity was able to pay for its daily caloric needs—as initial energy investment—and even add a capacity to the system of sustaining the breeding and reproduction of 7 more human individuals.

Finally, a frequent misconception regarding agrarian societies is the perception of a vertical structure regarding intra-social relations between classes for their total duration. Indeed, for their major duration, from early agriculture to late ancestry (10,000 BC–300 AC), agrarian societies were heavily dependent on human slavery for the construction of infrastructure and manual labor. However, in early medieval Western Europe and the Eastern Roman Empire (Byzantium), with the emergence of *feudalism*, relations between social classes were completely different. Transactions between nobles and farmers were based on a social covenant more akin to modern *land leasing* for the operation of agricultural works, monitoring output, productivity, and harvest optimization [47,48,51]. In this system, the nobles retained the *property* of the land and leased it to the peasantry via binding contracts of yield and productivity targets or quotas. These contracts included rights for the peasants to keep part of the output, while the major part was transferred to the owner as *land rents*. While, in ancient agrarian systems, the socio-ecological pyramid had a vertical structure even for the members of human societies, in medieval agrarian systems, this was horizontal, as depicted in Figure 4. A revival of slavery-based agrarian systems took place in the new world from 1492 to 1865; however, this can be considered as a historical repeat of the transition process from hunter–gatherers to agriculture in the new virgin

areas. The importance of socio-economic covenants in Europe became very clear during and after the bubonic plague (the "Black Death"), which spread between 1347 and 1353 and took the lives of around 75–200 million individuals. Following these years, peasants were such a rare resource that many of them were even able to negotiate the acquisition of property rights of the lands in which they were previously workers. This shift initiated their accumulation of wealth and ignited the beginnings of the bourgeoisie that steered the *Renaissance* and led to the introduction of the *Industrial Revolution* via financial capital investments regarding the accumulated agriculture-based surpluses in steam and internal combustion engines during the early and mid 1800s [33–35].

### *2.4. Energy Use Growth Macrodynamic Modeling*

Having presented the background and crucial elements of the agrarian energy paradigm between 10,000 BC and 1800 AC, we now describe the fundamental assumptions on which we structured the macrodynamic modeling as a dynamic function of the form:

$$x_t = f(x_{t-1}), \quad x \in R^+, t \in N^+ \tag{8}$$

Equation (8) depicts the general formulation of energy use growth macrodynamics as a discrete-time model that reproduces a causal self-feeding sequence. In parsimonious discrete-time population growth models, the fundamental parameters concern (a) an *intrinsic growth rate* and (b) a *carrying capacity*, as presented in Figure 3, forming a logistic growth pattern. Via a *Systems' Dynamics* approach [37], we graphically depict the growth dynamics between the two parameters, as shown in Figure 11, which presents the feedback loops as a circuit that completes a full feedback cycle after two successive (2) periods. Specifically, at a well-defined time step $t$ and for an initial carrying capacity $K$, any positive initial total population $x_0$ will grow by a positive constant rate $r$, which could be interpreted as the average number of offspring per individual. At time step $t$, the population growth will consume a fraction of the (assumed constant) carrying capacity. The abstraction of this fraction from $K$ will reduce the overall population growth at the next reproduction time step $t + 1$. This means that, although the average intrinsic growth rate remains constant ($=r$), the population has an intrinsic tendency to reduce its gross growth rate due to the consumption of the carrying capacity. An analytical mathematical formulation of Equation (8) and Figure 11 rationale is included in Appendix A for four (4) different dynamic maps.

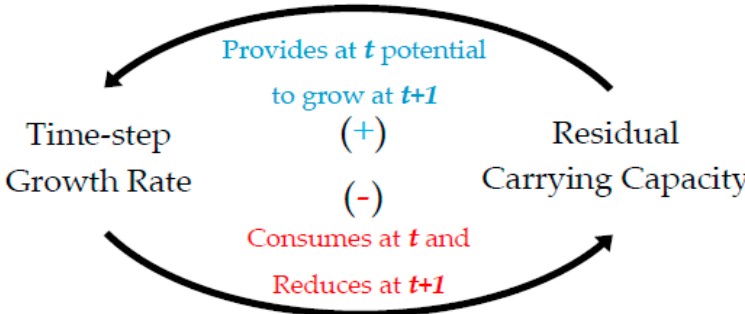

**Figure 11.** A systems' dynamics depiction of the feedback loops between the *intrinsic growth* and *carrying capacity* parameters in structured logistic growth dynamic models.

Moreover, discrete-time forms provide us with a variety of modeling and conceptual conveniences that are beyond the focus of continuous-time logistic growth and generally macrodynamic models [58]. For instance, the classical Verhulst or Gompertz logistic equations only reflect how the target variable's size relates to time as an exogenous variable without causal relations, only depicting the temporal map of its evolution. In addition, in continuous-time logistic growth models, systems always meet the carrying capacity as a global maximum size irrespective of their parameters' values without any fluctuations, chaotic behavior, or collapse. In contrast, even minimal logistic discrete-time models can

reflect a basic causality between the population sizes in $t-1$ and $t$ due to an endogenous feedback structure. Additionally, the parameters' values are of definitive importance for the system's evolution, potentiating a wide range of possible outcomes. Specifically, in addition to the features reflected in continuous-time models, they also parsimoniously incorporate the conditions under which the target population could stabilize smoothly, asymptotically, remain unstable but sustainable, become chaotic or become unsustainable, and collapse. In addition to its mathematical convenience, this reveals crucial economic aspects that are usually understated. As in natural ecosystems, *the agrarian paradigm's carrying capacity is renewable*, due to the constant replenishment of photosynthetic life covering the land by the practically *abundant solar energy.* This feature is missing from the *fossil-fueled paradigm*, which, although incomparable in terms of energy scale, is based on exhaustible deposits that would require a different modeling rationale with respect to carrying capacity and upper limits of utilizing available fuel stocks.

## 3. Results

In this section, we present the results of our simulation, which was carried out with the use of various data sources, to reconstruct a macrodynamic model of the global EEP growth from 10,000 BC to 1800 AC, taking into account the uncertainty of the (also reconstructed) raw data, as well as the differences in diets across the various geographical locations of the world, where, for simplicity, we had to assume a weighted average. As we also highlight in Appendix A, we diverted from pattern recognition econometric approaches on how an examined variable evolves as a function of time without the concern of charting causality topologies or feedback loops. Our approach focuses on the restatement and use of dynamic population growth models for very long periods of time (macrodynamics), testing their accuracy. Specifically, we examine (a) the empirical EEP via transforming reconstructed raw data and the ability of the four tested maps in Appendix A to reproduce its dynamics and (b) the concept of the *limiting factor* in relation to the *carrying capacity*, as a concept widely used in dynamic population growth models.

### 3.1. Energy Equivalent Population Growth Model Fits

The primary task carried out to implement the simulation involves transforming the raw data on the energy productivity of land into EEP. For this, we utilized the estimated land use ($m^2$) for the production of 1000 kcal nutritional value for a variety of foods [31,32]. As presented in Figure 12, estimations for 38 different food types concern their energy yields with modern production methods that are unlikely to accurately reflect the conditions and land use efficiency of the agrarian paradigm, which was probably much lower than today. However, as they comprise the best possible estimates, we may adopt them conventionally for our modeling purposes. In addition, an individual's daily caloric intake has probably varied significantly with various food combinations belonging to the available basket of Figure 12 or foods that are not even presented in the basket. The daily caloric intake has differentiated by geographical area, Koppen climate zone classification, altitude, etc., significantly impacting cultural and religious customs, as well as societal population sizes. Hence, due to this level of uncertainty and complexity, which does not even include spatiotemporal livestock composition, we chose to focus on the global scale, assuming caloric homogeneity. Based on these assumptions we estimated the weighted average of total agricultural land used for producing 1000 kcal of caloric value from each of the 38 food types. As the total land use is equal to 416.4580 $m^2$/1000 kcal, the weighted average is 74.0107 $m^2$/1000 kcal or 0.0074 ha/1000 kcal. In any case, the task of more accurately reconstructing the EEP via historical archives while including all possible nutritional combinations of food baskets spatiotemporally (i.e., at each historical period and at each geographical location) and including all types of livestock, following similar examples for more limited periods [48], comprises a major future research challenge.

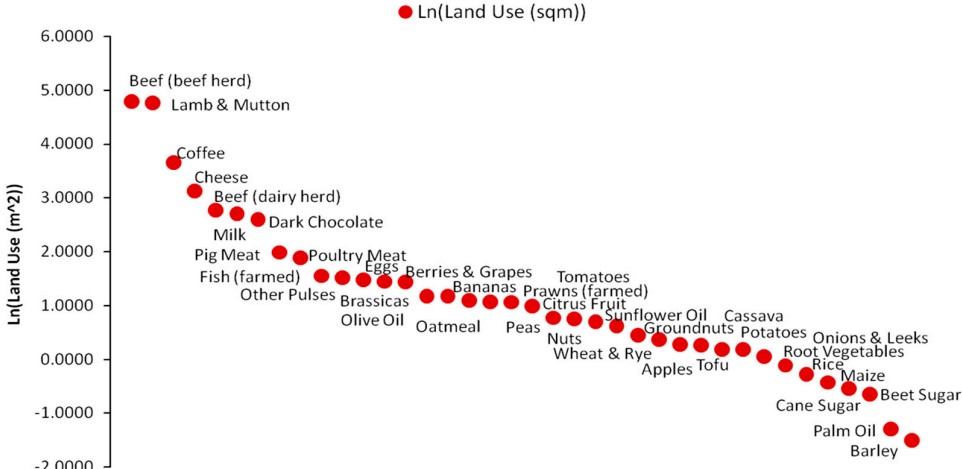

**Figure 12.** Intensity of land use (m$^2$) per 1000 kcal per type of produced food in descending natural logarithmic (Ln) scale.

Based on the data presented in Figure 10, we also assumed a minimum daily caloric intake of 3000 kcal per human individual. This level is 50% higher than the modern average recommended caloric intake of 1700–2000 kcal. However, considering all the heavy agricultural work of the average farmer, both in terms of manual and livestock coordination, which required almost all of his daily routine hours, and considering that the caloric intakes of individuals belonging in other social classes were significantly higher (by 125% for medieval aristocrats, by 175% for medieval monks, and by 200% for Roman legionnaires), this assumption is quite safe. By combining data on land use in the long term [29], land use of foods per 1000 kcal [31] and data on the global population growth [49] for the period 10,000 BC–1800 AC in Equations (1) and (2) in terms of yearly caloric needs (multiplying daily caloric needs by 365), we were able to estimate the EEP at each time step as the maximum number of individuals with yearly caloric needs of 1,095,000 kcal theoretically supported by total agricultural land.

The EEP for the major part of the examined period was estimated to be 60–80 times higher than the reconstructed human population. However, it should be taken into account that the total agricultural land energetically supported the global population of livestock either via grazing land or via the yield of cropland that was re-invested as livestock food. As presented in Figure 10, the individuals of some species like horses and cows required even 6–10 times higher caloric intakes than the reference farmer; pigs and sheep required caloric intakes near the human level, while poultry—mainly hens and chickens—required near 10 times lower daily caloric intakes. Furthermore, for such long intervals, we should consider the local, regional, and global events that disrupted agricultural yields along with the endogenous and systematic crop failures experienced in every agrarian society that are not accounted for. We express the econometric model as an objective function optimizing the estimated values of the predictor parameters *a,b* for each map at the natural logarithm scale (Ln) in terms of explained variance percentage (R$^2$) as:

$$\hat{a}; \hat{b} = g\left[ Max\left( R^2_{Inf(E_t)} \right) \middle| \hat{K}_t = K_{1800} \forall E_t \right], \quad a \in (1, +\infty), a > b, b, E \in R^+ \tag{9}$$

Equation (9) suggests that the optimal values of parameters *a,b* are a function of the R$^2$ value maximization, constrained by an upper EEP size equal to the EEP of 1800 AC, as well as positive values for parameters *a,b*, and the EEP. The observed EEP reproductions by each model are presented in Figure 13.

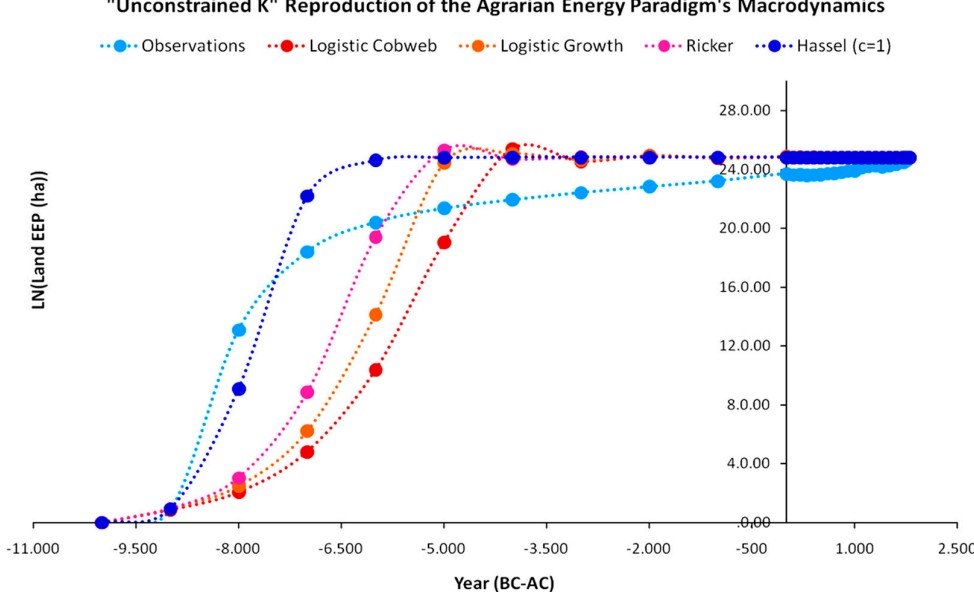

**Figure 13.** Reproduction of EEP macrodynamics according to the transformation of reconstructed raw data and the simulations based on the four dynamic maps (Logistic Cobweb, Logistic Growth, Beverton–Holt, Ricker) for the unconstrained $K_{1800}$ model version.

Regarding carrying capacity, the year 1800 was chosen for the first model version as the year of maximum global energy yield (in kcal) derived *exclusively from agricultural practices* as it is the last pre-industrial year of the agrarian paradigm in which agricultural practices were dominant (before the large-scale use of petrochemical fertilizers). This is a rather lenient assumption as it is highly unlikely that early agrarian societies in the years 10,000 BC or 1000 BC had any idea of the global carrying capacity of the systems in 1800 AC. In any case, for analytical purposes, we examined this model version as the "unconstrained $K_{1800}$" assumption.

As presented in Figure 13, although the performance of all models is high, with $R^2$ values higher than 0.89, the Beverton–Holt (Hassel $c = 1$) performs best. In addition to its higher $R^2$ value, it reproduced the empirical EEP growth at its initial stages much more accurately than the other models. While all other models reproduce EEP at a much lower growth rate and minimize the deviations from 6500 to 5000 BC, the Beverton–Holt model is able to minimize deviations much earlier (from year 8000 BC), with its residuals being more uniformly distributed. The optimal parameter values and $R^2$ performances for each of the maps of the "unconstrained $K_{1800}$" model version are presented in Table 1.

**Table 1.** Parameter *a,b* values of the EEP simulation dynamics with the four maps along with the $R^2$ values of the parameters' regression for the unconstrained $K_{1800}$ model version.

| Population Map | Parameter *a* | Parameter *b* | $R^2$ |
|---|---|---|---|
| Logistic Cobweb | 2.4354 | 0.0578 | 0.8924 |
| Logistic Growth | 1.6931 | 0.0681 | 0.9115 |
| Beverton–Holt (Hassel *c* = 1) | 14.6546 | 0.5499 | 0.9752 |
| Ricker | 3.3850 | 0.0491 | 0.9338 |

However, although all four models demonstrate high performance in terms of $R^2$, all fail to accurately reproduce the rate at which the empirical data approach carrying capacity. While the empirical EEP constantly grew until 1800 AC at a diminishing rate, all models arrive at the 1800 AC carrying capacity earlier, always staying above it with small oscillations until the empirical EEP eventually converges. After the year 5000 BC,

all four models will asymptotically converge to the carrying capacity and remain stable at it until 1800 AC. The major argument in favor of this approach is that this carrying capacity level was feasible, and the agrarian civilization was consuming it at a very low rate. However, a possible counter-argument against this is that it could be considered unrealistic as, for thousands of years before 1800 AC, the carrying capacity level was probably much lower, especially with lack of technological upgrades and agricultural methods that appeared much later. In turn, if we accept that the carrying capacity was much lower at the beginnings of the agrarian energy paradigm, although, mathematically, the residuals are distributed both below and above the simulation models, it would have been impossible for the agrarian civilization (as the sum of all agrarian societies in the Earth) to be operating above its carrying capacity for such a long period. It would be theoretically possible for a number of years and even decades; however, for such long time steps of 1000 years each, this assumption is practically unacceptable.

A possible refinement for this issue could be to assume a different carrying capacity at each time step with a changing upper limit, that is, to substitute $K_{1800}$ (assumed to be valid from 10,000 BC) with a carrying capacity as a function of each time step *K(t)*. In this case, *K(t)* would be equal to the maximum energy yield by total agricultural land at each time step, as a more realistic representation. However, as the time steps for the available data are intervals of 1000 years each for the first 10,000 years, such an assumption would suffer from the exact same issues and require the assumption of variable values of parameters *a,b*, from which the carrying capacity emerges. This would increase the model's complexity without offering substantial value to its predictive ability. Instead, to preserve parsimony we tested the effect of an additional simple constraint regarding the carrying capacity. Specifically, we assumed that, at each time step *t*, the simulated EEP size should not exceed the empirical EEP size. By intensifying this constraint, we can re-write Equation (9) so that it resembles the following form:

$$\hat{a}; \hat{b} = g\left[Max\left(R^2_{Inf(E_t)}\right) | \hat{K}_t \leq K_t \forall E_t\right], \quad a \in (1, +\infty), a > b, b, E \in R^+ \tag{10}$$

With the above re-postulation of the optimization function, Equation (10) indirectly incorporates the assumption that, for the very long time intervals between 10,000 BC and 0, the carrying capacities were at every time step both variable and not overshot by the observed EEP. With this input, we examine the new optimal values for parameters *a,b* along with their $R^2$ performance, considering them constant for 10,000 BC–1800 AC.

The results of the "constrained *K*" version with respect to how the models perform with the intensification of the constraint on carrying capacity are presented in Figure 14. As the new constraint forbids the simulated EEP from exceeding the observed (reconstructed) EEP at every time step *t*, the next optimal solution is found for simulated EEPs at lower levels. Indeed, in this version, at each time step, all models yield EEP levels below the observed EEP, as they cannot reproduce it with full accuracy. Specifically, once again the Beverton–Holt model prevailed in terms of $R^2$ and once more demonstrated the best performance in reproducing accurately the EEP growth at the initial time steps of the agrarian paradigm, while the other three models started converging to the observed EEP only after 5000 BC. In terms of $R^2$, the changes for the logistic cobweb map and the logistic growth map were 4% and 6.1% reduction, respectively, while for the Ricker map, this figure was 0.7%, and for the Beverton–Holt map, this figure was just 1.1%. Considering that the constraint's intensification provides a more realistic state of the EEP growth dynamics, the reduced $R^2$ values could be considered insignificant, especially for the Ricker and Beverton–Holt maps that embody a number of interesting economic interpretations regarding intra-social competition.

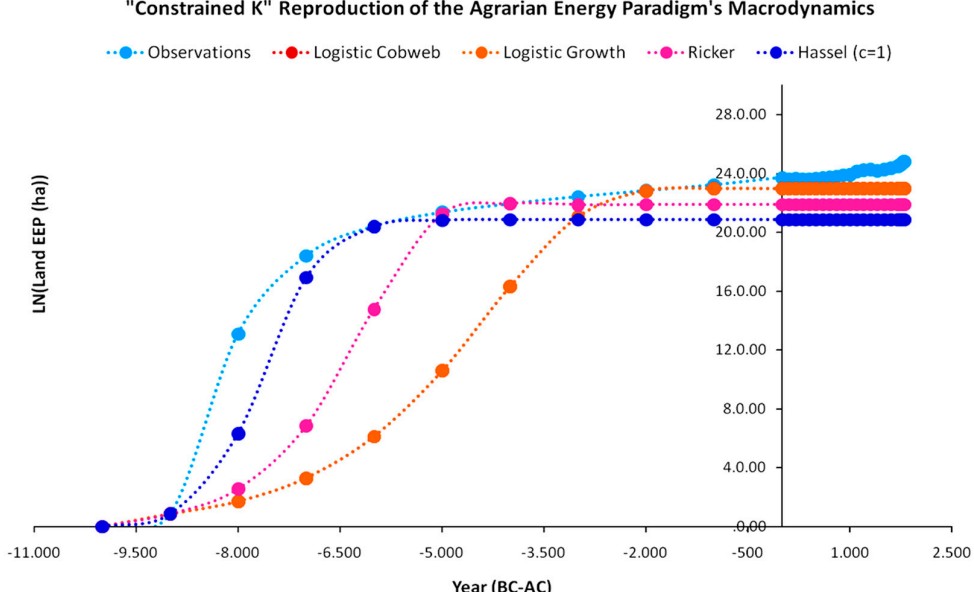

**Figure 14.** Reproduction of EEP macrodynamics according to the transformation of reconstructed raw data and the simulations based on the four dynamic maps (Logistic Cobweb, Logistic Growth, Beverton–Holt, Ricker) for the constrained *K* model version.

Regarding the new optimal parameter values, for the logistic cobweb and the logistic growth maps, we can intuitively expect the constraint imposed on the model to not exceed the carrying capacity at any time step, to set parameter *a* at the optimal value leading to the OSMP, as shown in Equations (A6), (A7) and (A13), as well as Figure A1 of the Appendix A. With this value of parameter *a*, the optimization exclusively depends on the parameter *b* value. Hence, for the logistic cobweb with *a* = 2 and the logistic growth map with *a* = 1, the optimal value of parameter *b* is the same for *b* = 0.0435, meaning that they completely overlap at every time step. A different state applies for the Ricker model with value *a* = 3.0599 above its OSMP value *a* = 2.71, as proven in Appendix A. The Ricker map is the only map exhibiting a very slight oscillation between 4000 BC and 0; however, this is within the limits of the constraint. The new parameter values and $R^2$ performances are presented in Table 2.

**Table 2.** Parameter *a*,*b* values of the EEP simulation dynamics with the four maps along with the $R^2$ values of the parameters' regression for the constrained *K* model version.

| Population Map | Parameter *a* | Parameter *b* | $R^2$ |
|---|---|---|---|
| Logistic Cobweb | 2.0000 | 0.0435 | 0.8559 |
| Logistic Growth | 1.0000 | 0.0435 | 0.8559 |
| Beverton–Holt (Hassel *c* = 1) | 9.9311 | 0.4281 | 0.9741 |
| Ricker | 3.0599 | 0.0511 | 0.9271 |

However, there is a significant difference in relation to the $K_{1800}$ version. Although the Beverton–Holt map prevails even in this model version, due to the constraint's intensification, it does not converge with all other maps in the carrying capacity but stabilizes at a level that is the lowest in relation to all other maps. Although the Ricker map converges to the observed EEP later than the Beverton–Holt map, increasing the deviation, it compensates for this by reducing the distance from the carrying capacity *K* at year 1800. The logistic cobweb and logistic growth maps both stabilize even closer to the carrying capacity of the year 1800. However, their distances from the observed EEP are so high at initial stages that even their better approximation after 2000 BC is unable to compensate.

Regarding the economic meaning behind the maps' performances, the prevalence of the Beverton–Holt map may be an indication of *contest competition* [59], which is directly

related to rigid and well-founded socio-ecological hierarchy in space and time. Contrarily, the Ricker map, as the second optimal map, embodies properties of *scrambled competition*, which, in this particular case, along with the specific parameter values, remains very slightly manifested (though oscillating it leads to the convergence and subsequent stabilization of the carrying capacity). This could also be considered as an additional indication of contest competition in the macrodynamic view. As explained in Appendix A, from a socio-ecological view, a full-scale manifestation of scrambled competition would lead to continuous conflict over land-derived energy resources and a type of social anarchy that would suggest a continuous bouncing between high and low levels of energy availability. In turn, this would be equivalent to maximum entropy state constrained by always positive EEP levels, thus securing a minimum population above thermodynamic equilibrium (=0), as suggested by the heavy-tailed Ricker map.

In any case, we should consider that, even if contest competition prevails in a society, it coexists with elements of scrambled competition at various intensities. Even if agrarian societies were organized as socio-ecological pyramids for the allocation and utilization of secondary solar energy stored in agricultural land—in the role of the *energy currency*—a statistically significant fraction of the global population historically resorted to practices of rogue resource harvesting such as bandit thievery, piracy, etc. In addition, although all agrarian societies had some level of hierarchical organization, as evidenced by their ordered and unequal distribution of their energy budget, at a higher analytical scale, societies often engaged in wars, which equates to lower-intensity scrambled competition. Irrespective of the energy expenditure during the conflict, the winner would incorporate the new lands into its territory and restore contest competition on its own terms, either by replicating it in the case of full land absorption or establishing a rent paid by the conquered culture and leaving the previous contest competition practices relatively intact. This was frequently observed in the case of multinational empires, such as the Western and Eastern Roman Empires, which had to rule with reasonable fairness over many heterogeneous cultures. In short, while data indicate that contest competition was prevalent in intra-social relations, scramble competition was observed in inter-social ones.

### 3.2. Carrying Capacity Mechanics: The Limiting Factor

A critical dimension of natural resource economics, observed in every energy paradigm, concerns the formation mechanics of the *carrying capacity K*. In the majority of logistic growth models (continuous and discrete time), the carrying capacity concept suggests the existence of a thermodynamically finite pool of resources from which the latter are drawn and later combined to produce a final product. However, among a high variety of natural resources with different physical properties, availabilities, along with available scientific knowledge and applied technology that determines their costs and substitutability, it is usually technically difficult to quantitatively estimate the maximum potential use of each one of them and come up with a carrying capacity scalar index. To resolve this issue with parsimony and consistency, we adopt the *Liebig–Sprengel Law of the Minimum*, which, in turn, establishes the *Limiting Factor* concept [60]. Although this law can be generalized to apply to all energy paradigms, it is of particular importance for the agrarian energy paradigm, as the law was designed to initially focus on agricultural ecosystems, stating *that plant biomass growth is limited by the nutrient with the least natural availability*. The original postulation of the law concerned the ratios at which *Carbon* (*C*), *Nitrogen* (*N*), and *Phosphorus* (*P*) are combined to form plant biomass with the empirical relationship (*L*):

$$L_{C:N:P} = (C/N/P) = (41/7/1), \quad (C/N/P) \in R^+ \tag{11}$$

According to Equation (11), the three elements have to be combined in this specific ratio to form one unit of biomass, meaning that, for the plant to optimize the utilization of nutrients and maximize its biomass and energy intake, the natural availabilities of these nutrients should be found *exactly* at the same ratio as their demand from the plant. More specifically, the deposits of carbon, nitrogen, and phosphorus in the environment should

be the *exact multiples* of the plant's demand with no *residuals* (z); hence, for every 1 part of phosphorus, there should be 7 naturally available parts of nitrogen and 41 parts of carbon. We can mathematically formulate the identification of a limiting factor's existence as a *Greatest Common Divisor* (*GCD*) target of natural stocks ($K_i$) as follows:

$$LF_t \exists \forall (C/N/P) \neq \left[ \frac{(K_C/K_N/K_P)}{GCD_{C,N,P}} | z = 0 \right], \quad GCD_{C,N,P}, z \in (0, +\infty) \tag{12}$$

Equation (12) suggests that, if there exists a *GCD* to ensure that the ratios of natural stocks of nutrients in the environment are an *exact* multiple of the ratio of nutrients' demand from the plant, *no limiting factor exists*. Contrarily, for any deviation from this optimal state (no GCD found or $z \neq 0$), *a limiting factor exists*. In its general mathematical form, for every energy paradigm in humanity's history and for every final product that combines at least 2 species ($i > 2$, with $i \in N^+$) of natural resources $X_i$ with each species having a confirmed amount of natural deposits $Y_i$, we may reformulate Equation (12) as:

$$LF_t \exists \forall (X_1/X_2/ \dots /X_n) \neq \left[ \frac{(Y_1/Y_2/ \dots /Y_n)}{GCD_{1 \to n}} | z = 0 \right], \quad Y_{i \forall i \in [1,n]}, GCD_{1 \to n}, z \in (0, +\infty) \tag{13}$$

Although Equation (13) sets the conditions of a limiting factor's existence, we need to formulate the conditions for the identification of which is the system's limiting factor. In terms of demand to availability, we may write that the limiting factor satisfies:

$$LF_t = Max[(41/K_C); (7/K_N); (1/K_P)], \quad K_{C,N,P} \in R \tag{14}$$

The equivalent formulation of Equation (14) expresses the resource with the fastest natural stock depletion ($K_i$) for fixed demand at each time step $t$:

$$LF_t = Min[(K_C/41); (K_N/7); (K_P/1)], \quad K_{C,N,P} \in R \tag{15}$$

Based on Equation (15), the limiting factor is simply the nutrient that will be depleted faster than any other nutrient (at a current consumption rate). Essentially, Equation (15) is a special expression *Reserves to Consumption* ratio (*R/C*) that yields the remaining time of a resource's use. The same rationale may be followed for generalizing Equations (14) and (15) for any natural resource species and its stock, as in Equation (13).

In direct relation to the limiting factor concept, we may examine briefly indicative data for other resources used, in combination to land use presented in Figure 12 for the production of 1000 kcal value. For instance, the freshwater withdrawals for the production of 35 foods are presented in Figure 15 [61]. Assuming that land use and freshwater withdrawals are optimally combined to produce each 1000 kcal portion per food type, we may consider the effect of misusing other complementary resources, such as fertilizers and nutrients. Such an example is presented in Figure 16 in the form of eutrophying $CO_2$ emissions caused by the excessive (non-optimal) use of fertilizers as PO4 equivalents for a yield of 1000 kcal and for the same basket of 35 food types as for freshwater withdrawals. PO4 constitutes the energy currency for various biomass metabolic processes [62]. The highlight in Figure 16 in relation to the set of Equations (11)–(15) is that if a limiting factor exists (meaning that there will be a surplus in the natural availability of other resources combined to it), any additional intake from the other resources is unable to compensate for the deficit of the limiting factor's natural availability. From an economic standpoint, such a state establishes a *perfect complementarity* between these resources, hence their *zero substitutability*. Essentially, *in terms of natural availability, a physical trade-off between resources in excess and resources in deficit is impossible*. Moreover, in agriculture, an excess input of a resource with higher natural availability will fail to be metabolized by plant biomass and will be transferred via various water flow paths to other ecosystems causing eutrophication and other pollution impacts.

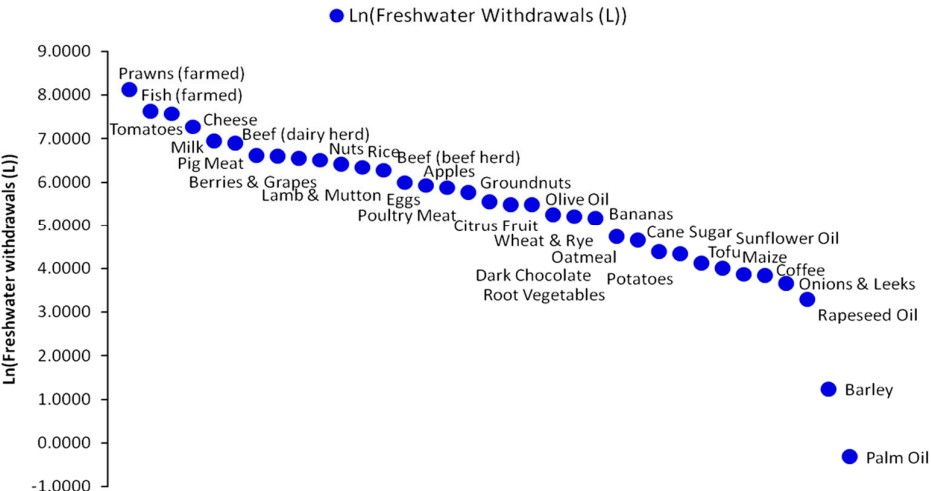

**Figure 15.** Intensity of freshwater withdrawals (L) per 1000 kcal per type of produced food in descending natural logarithmic (Ln) scale.

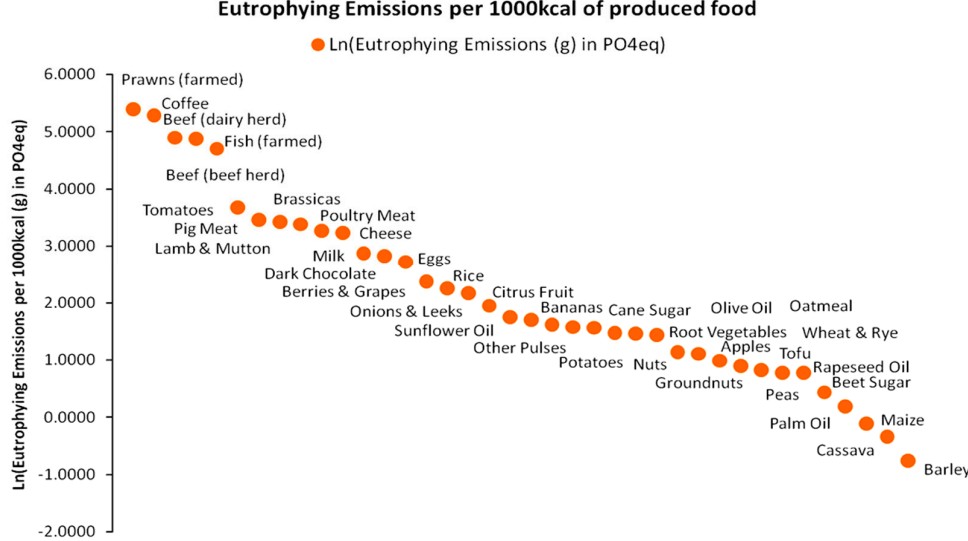

**Figure 16.** Intensity of eutrophying CO2 emissions (g in PO4eq.) per 1000 kcal per type of produced food in descending natural logarithmic (Ln) scale.

The state of perfect resources' complementarity has provided a motivation for the postulation of numerous agricultural production functions that are widely used in both subsistence and industrial agricultural profiles [62], with many of them having properties that suitably fit the agrarian energy paradigm [63,64]. Equation (13) and the perfect complementarity condition may not apply universally, as many natural resources can be used alternatively (e.g., metals) up to a state of *perfect substitutability*, where full trade-offs can take place [10]. For agro-economic systems though, we can safely assume that the perfect complementarity condition applies. In regard to the agrarian paradigm, significant geographical variability in limiting factors across the world should also be assumed and not only for the three major nutrients presented above (e.g., in desert ecosystems, the limiting factor tends to be Sulfur). Furthermore, temporal shifts in limiting factors across the improvement in sowing, tillage, irrigation, and harvesting methods that are the equivalent to technological upgrades affecting *the Water–Energy–Food* nexus should be considered more than likely, although, at the examined spatiotemporal scale, they have an insignificant effect on macrodynamic modeling and the available datasets used for it.

Finally, another facet of the limiting factor concept identified in our work concerns the nutritional combinations of humans and livestock. However, as the caloric intake exclusively concerns the *quantitative* part, ignoring the *qualitative* part, similarly to the soil, we could assume that a complete nutrition consists of proteins, vitamins, hydrocarbons, and fats from food combinations out of a total basket composed of the foods presented in Figures 12, 15 and 16, with a minimum intake amount from each. In this context, these baskets are essentially subset elements of a total food "pool", forming a number of qualitatively equivalent combinations, with *substitutability* between them. Incorporating this aspect would skyrocket the complexity of our work. However, we argue that the EEPs of agrarian societies lacking in such combinations would be more limited (the small society in New Guinea examined in Section 2.1 and Figure 5 could demonstrate such a case).

## 4. Discussion

In this section, we discuss two significant aspects of our work that were incorporated into the core assumptions; however, a thorough examination of these aspects is out of the scope of this paper. Specifically, as major extensions of our work, we discuss (a) the issue of the *Energy Paradigm Scale*, further substantiating the utilization of the 1800 AC land use level as a carrying capacity benchmark and (b) the issue of *intra-social competition* and its effect on system stability in the context of the growth maps developed in Appendix A.

### 4.1. Energy Paradigm Scale

Societies that distribute a large fraction of their surpluses to technological upgrades benefit from a higher potential and an extended time of use. This is directly related to the *second law of thermodynamics*, as, whatever the theoretical energy use potential may be, the thermodynamic efficiency $h$ will always be between 0 and 100% [$h \in (0, 1)$]. In this context, the central issue is the optimal rate between the energy use growth and energy efficiency increase over time. Technological upgrades increase the *useful/non-useful* energy ratio introduced to the economic system, maximizing useful work and providing it with more degrees of freedom to create sophisticated structures. On the other hand, economies that rapidly boost their energy use, consume their resource reserves faster with less efficient technology, and although they may experience an artificial state of "energy plenty", they find themselves locked in a lower energy potential growth path than if they had systematically re-invested a part of their surpluses in technical upgrades. Irrespective of the examined energy paradigm, the role of technological upgrades, whether they concern more efficient methods to combine better cropland and grazing land or increase the fraction of thermomechanical work to thermal losses in an internal combustion engine, is vital for both the scale and sustainability of society's energy use.

Energy availability in human economies and technical improvements mitigating the pressures of limiting factors has been the cornerstone of economic wealth augmentation, knowledge accumulation, and structural complexity across all the stages of social evolution [5–7,9,10,22,24–27,33–35], from hunter–gatherer groups, their transition to agrarian societies, and their transition to the industrial civilization. The scale of energy use with the availability of modern technology is literally incomparable to the one of the agrarian paradigm, as revealed from the skyrocketing of related *energy density* data. Indicatively, in regard to the power generation density, scholars suggest [33–35] that phytomass has significant power generation variability (highly affected by the high diversity of plant species) ranging from 0.1–10 W/m² for a respective land use range between 1 m² and 1 km², meaning that phytomass power density ranges from $10^{-4}$ W/m² to 1 W/m². With estimations from the same source and with the same rationale, the power density of a standard thermal power plant ranges from $10^{-3}$ W/m² to 10 W/m², setting an order of magnitude of 10 for the lowest and highest power densities—excluding all qualitative differences concerning internal combustion technology that was completely unavailable in agrarian societies. This unprecedented increase in power density can practically be interpreted as the equivalent to the *mitigation of the land's limiting factor upon energy generation*,

releasing it for other uses such as large-scale petrochemical-based agriculture. Additionally, these estimations are also supported by the respective growth of global population [49], as, from a total of 1 billion people near the end of the agrarian paradigm (1800 AC), we observe an eight-fold growth today, following the increase in land use carrying capacity.

In direct relation to the large-scale introduction of fossil fuels in human societies, the energy state of agriculture at each energy paradigm has been a subject of discussion for numerous researchers [11–16,23,30,39,40,47,48]. The core distinguishing feature though is what identifies the phase change, i.e., the shift from agrarian societies to industrial societies. With the Industrial Revolution, modern agriculture practically transforms *from a net energy supplier to a net energy user* [54] via the extensive use of fossil fuels (that substitute solar energy inputs) and petroleum derivative products, mainly petrochemical fertilizers. Although industrial agriculture has skyrocketed the food productivity of land it would be unable to achieve it without detaching agriculture from its previous long-term interlocked solar energy flows. It is not an exaggeration to say that *petrochemical fertilizers are the land's analog of artificial steroids in humans*. The cost of this phase change consists of the large-scale environmental impacts [36–38] that require alternating fallow periods for the lands to replenish the minimum level of their nutrients from natural processes. This approach of agricultural energetics comprises a compass for economies that are currently in the phase of agricultural subsistence, growth, and diversification that empirically precedes the phase of industrialization [54,55].

*4.2. Competition and Stability*

Aside from the energy use scale as an aggregate index of a society's energy availability, a less discussed aspect concerns the impact of the internal structure of society on its energy use level, growth and sustainability in the long-run. As presented in Appendix A, the amount of energy use does not by itself constitute a condition for ensuring the system's sustainability, unless accompanied by an efficient mechanism of technology transition. This conclusion directed contemporary social research to the study of endogenous factors of a system's energy evolution, as well as the detection and remedy of its structural fallacies that would potentially threaten it with collapse [4]. It has been substantiated both theoretically and quantitatively that social hierarchy and stratification [22] are endogenous features that allow for the allocation of wealth in human societies. Our work shares and expands upon these views, arguing that the universal currency of wealth is the available thermomechanical work for building up social structure and complexity.

As presented in Appendix A, in the properties of the four maps, a rapid energy use growth via an excessively high intrinsic growth rate for a constant carrying capacity increases the probability of the chaotic evolution of a society's energy ecodynamics, which may eventually lead to structural decomposition, corresponding to White and Tainter's arguments for the collapse of the Western Roman Empire. Although intra-social competition is more substantiated for the Hassel family, with *contest competition* suggesting intensive social hierarchy and population control in the Beverton–Holt version and *scramble competition* suggesting a more uniform social structure in the Ricker version, intra-social competition in the logistic cobweb and logistic growth models remains quite obscure, as they embody elements of both competition types. We may argue that the instability phase in these two maps reflects a special case of a combination of low intra-social competition and collective agreement on aggressive patterns of energy harvesting from the environment. Such a pattern may be manifested in various ways, such as geographical expansion via waging war or locust-type resource consumption and nomadic relocation.

The Hassel and Ricker maps provide many conceptual insights into the above. In the monotonic Beverton–Holt model, hierarchy is so intense to the point where practically eliminates the risk of resource insufficiency for the members of society via maintaining strict population control by reproducing itself at the probable cost of lack of opportunities to climb the hierarchy ladder from individuals residing at lower levels. Contrarily, in the Ricker model, intra-social competition expands across the social hierarchy levels and is

so intense to the point where the resources become frequently insufficient to cover the minimum needs of every individual of the population, sowing endogenous instability. The heavy-tailed exponential map though also signifies that, while the population could fall to extremely low (near extinction) levels, its asymmetry implies the population's resilience to remain positive and recover. A possible economic interpretation of this feature would be that, after a severe period of social structure decomposition and population size reduction, available resources can compensate for the minimum available needs of each individual while the remaining population itself develops synergistic behavior, leading to restored social complexity. Indeed, as observed [4], the collapse of the Western Roman Empire was followed by economic rejuvenation, with the new simpler and more decentralized social structures forming the basis for the development of feudal systems in the Middle Ages. In contrast, both logistic maps lack this property of the Ricker map, where very high intrinsic growth ratios will not only cause the population to intensively fluctuate but also put it in an unsustainable growth path and considerably increase the risk of its permanent extinction (for $r > 4$ as shown in Appendix A).

## 5. Conclusions

Our work develops a theoretical framework that can be used to depict human civilizations as evolutionary metabolic ecosystems where energy use has a fundamental role for their structural sustainability and diversification [56]. The application of quantitative methods for examining the historical course of civilizations across large spatio-temporal scales is the core of the field of *macrodynamics* that has been adopted in this study. The first pillar of our work concerns how the conceptual framework can provide insight into how civilizations operate as ecosystems in space and time. The core concept is the *Energy Paradigm*, defined as the dominant pattern of energy harvesting from nature. The energy paradigm concept is vitally important to achieving a minimum level of homogeneity and comparability between different periods of human civilizations. By defining the different energy paradigms, we practically define the periods of fundamental *structural changes in energy harvesting patterns*. Hence, according to the above argumentation, we may outline three sequential types of socio-ecological organization with their dominant energy harvesting pattern as criterion: the *hunters-gatherers*, the *agrarian societies*, and the *fossil-fueled industrial societies*. Our work focuses on the study of the agrarian paradigm for the period 10,000 BC–1800 AC as this is the period in which humans first achieved the large-scale and systematic accumulation of net food surpluses by harnessing secondary solar energy inputs in the form of plant biochemical energy. This process, in turn, triggered a series of unprecedented changes on a technical level via the leveraging of the power of domesticated animals and the relationships between the members of societies regarding the allocation of produced wealth.

The second pillar of our work concerns the presentation of empirical evidence for the examined period by composing and correlating reconstructed data from various databases and sources on energy use. Such data concern the estimated daily caloric needs of various animals and humans per social class at various periods of the agrarian paradigm, the growth of cropland and the introduction of grazing land that supported the domesticated animals' population for leveraging their muscle power to shift agricultural output from the subsistence state to the surplus state. In regard to the relation between cropland and grazing land, we examined the evolution of their Fisher–Pry ratio, as partially complementary and partially competitive technologies, accompanied by a set of correlations between global population growth and the constituents of total agricultural land. In this pillar, we also examined the impact of intra-social complexity on energy use sustainability. For this argument, we began by utilizing the theoretical findings of Leslie White and Joseph Tainter on the collapse of the Western Roman Empire as the most indicative and well-recorded case of the agrarian paradigm. In more detail, the findings of these scholars essentially suggest that although energy availability comprises the *necessary condition* for the sustainability of a complex society, it does not, by itself, comprise a *sufficient condition*, as the collapse of

the Western Roman Empire shows. To support these historical studies quantitatively, we developed a mathematical framework of logistic growth based on four different discrete-time dynamic maps with two parameters, namely intrinsic growth and carrying capacity. In this context, we discussed the effect of the parameters' values on the maps' stability, oscillation, or collapse as well as their economic interpretations.

In the third pillar of our work, we performed a simulation and examined the results of the four models based on the respective four maps. As we based our analysis on reconstructed data that already embody high spatio-temporal uncertainty with respect to human and animal diets, carrying capacities and agricultural disruption events—such as extreme weather, famine and failed crops—we presented thoroughly our assumptions for transforming the available data to *Energy Equivalent Population* (EEP). To implement this as accurately as possible, we utilized data on the productivity of land per 1000 kcal of nutritional value of various crops, and adopted weighted average sizes. The estimation of the global EEP was a challenging task that involved incorporating numerous significant uncertainties, although for a small-scale agrarian society in New Guinea, the results were very accurate. After the generally quite satisfactory performance of all models, with the Beverton–Holt model prevailing in terms of explained variance, we discussed the crucial aspects of logistic growth models, such as the emergence of the widely used concept of *carrying capacity* from the parsimonious and mathematically elaborate concept of the *limiting factor* (along with its origins, modeling, and generalization), and the role of *competition* in the growth of a population.

Taking into consideration the large uncertainties mentioned in our empirical work, this paper acts as a starting point for advancing the research questions and elaborating our quantitative analysis. The aims of our future research targets include the following: (1) to contribute to *more accurate data reconstructions* on energy use in the agrarian paradigm and (2) to *downscale our econometric analysis* to the micro-dynamic level, where we can study the structural elements of socio-ecological pyramids in Figure 4 for each continent and extrapolate their possible relations via trade. Currently, we are examining the data reported by Gilbert at al. [65] in our attempt to restructure and estimate the populations and EEPs of the various livestock types as a function of cropland and grazing land using HYDE 3.2 data [30] for the period 10,000 BC–1800 AC as accurately as possible. In this way, we will be able to elaborate our models by (a) estimating the diversity [66] of the regional and global socio-ecological pyramids in relation to Equation (6) in regard to the equiprobability of its various levels, as only normalized EEP sizes should be examined to reveal true energy stratification in agrarian societies, and (b) apply panel-data econometric methods [67] to perform Granger causality analyses [68] between the structure of socio-ecological pyramids and their spatio-temporal (i.e., by continent and by period) sustainability in relation to their limiting factors and environmental footprints.

**Author Contributions:** Conceptualization, analytical framework, methodology, G.K.; formal analysis and writing, G.K. and N.M.; visualization, G.K.; manuscript review and refinement, G.K. and N.M.; research, G.K.; manuscript supervision and quality control, N.M. All authors have read and agreed to the published version of the manuscript.

**Funding:** The authors received no funding for their research and the writing of the manuscript.

**Data Availability Statement:** All raw data used for this work are publicly available and are referenced in the "References" section. Processed data will be provided by the authors upon request.

**Acknowledgments:** The authors would like to express their appreciation for the help and support they received from the MDPI Land Editorial Office and the Special Issue Guest Editors.

**Conflicts of Interest:** The authors declare no conflict of interest.

## Appendix A. Energy Equivalent Population Maps

In this part, we theoretically examine the mathematical and graphical features of the four logistic growth *maps* that were examined to reproduce the EEP ($E_t$) across the agrarian energy paradigm. A map is typically a full set of points that host all possible population growth paths based on Equation (8) for an infinite range of initial values for a fixed set of parameters *a* and *b*. Each of the examined maps consists of three structural elements: *Growth Rate Plot*, *Cobweb Plot*, and *Temporal Population Growth Plot*.

*Appendix A.1. The Logistic Cobweb Map*

We may reformulate Equation (8) in the form of a simple density dependent logistic growth model of two parameters: one that depicts the *intrinsic growth ratio* of the population (its natural tendency to grow without necessary knowledge of its natural limitations) and one depicting its *carrying capacity* [69]:

$$x_t = a \cdot x_{t-1} \cdot \left(1 - \frac{b}{a} \cdot x_{t-1}\right), \quad a > b, a, b \in R^+ \tag{A1}$$

In Equation (A1), parameter *a* depicts the *intrinsic growth ratio*, and parameter *b* is the *resource efficiency coefficient* or *population limitation intensity coefficient*, deriving from the impact of consuming the carrying capacity. Another significant mathematical aspect of the formulation of Equation (A1) is its use either as a *difference* or as *growth* equation. In our work, unless stated otherwise, we use it as a growth equation by simply suggesting that *b/a* = 1/K used in standard r-K models.

Specifically, at a well-defined time step *t*, for an initial carrying capacity, any individual of any positive initial total population *x* will grow by a rate of *a*, which could also be the average number of offspring per individual. The population growth will consume a fraction of the (assumed constant) carrying capacity. This consumption (at the next reproduction time step *t + 1*) will have a negative impact on the overall population growth by a factor of *b*. This means that although the average intrinsic growth rate remains constant (=*a*), the population has an intrinsic tendency to reduce its gross growth rate due to the consumption of carrying capacity. This may occur via a higher number of deaths in the population or via intense competition and the selection of offspring to be protected by the parents. How this will occur internally in the population's society is not depicted in such models, but its overall effect is included via parameter *b*. Using a stochastic framework [59], many authors [59,70,71] have thoroughly discussed the various interactions between a population's individuals to accurately depict internal competition patterns. As these extensions hold economic meaning for the examined models, we thoroughly discuss their aspects concerning the growth maps in Section 4.2 on competition and stability. Through using this approach, the carrying capacity *K* is actually the ratio of parameters *a/b*, leading to the following:

$$K = a/b, \quad K \in (1, +\infty) \tag{A2}$$

The first thing that is significant to note here is that parameter *b* expresses the *intensity* of *gross* population reduction in relation to parameter *a*, as shown in Equation (A1), so that a temporary *net exponential population growth* is not prohibitive in the model. Although, in general, the logistic growth pattern primarily depends on the initial population and the ratio between parameters *a* and *b*, in general, a low carrying capacity consumption impact intensity (low value of parameter *b*) will, in the initial stages, allow for exponential growth in the population. Only at later stages, where residual carrying capacity becomes intensively scarce, will the net population growth start diminishing until it reaches zero, balancing at the steady-state of exact population replenishment (irrespective of its internal structure).

According to the above mathematical context, the *Logistic Cobweb Map* dynamics are graphically presented in Figure A1.

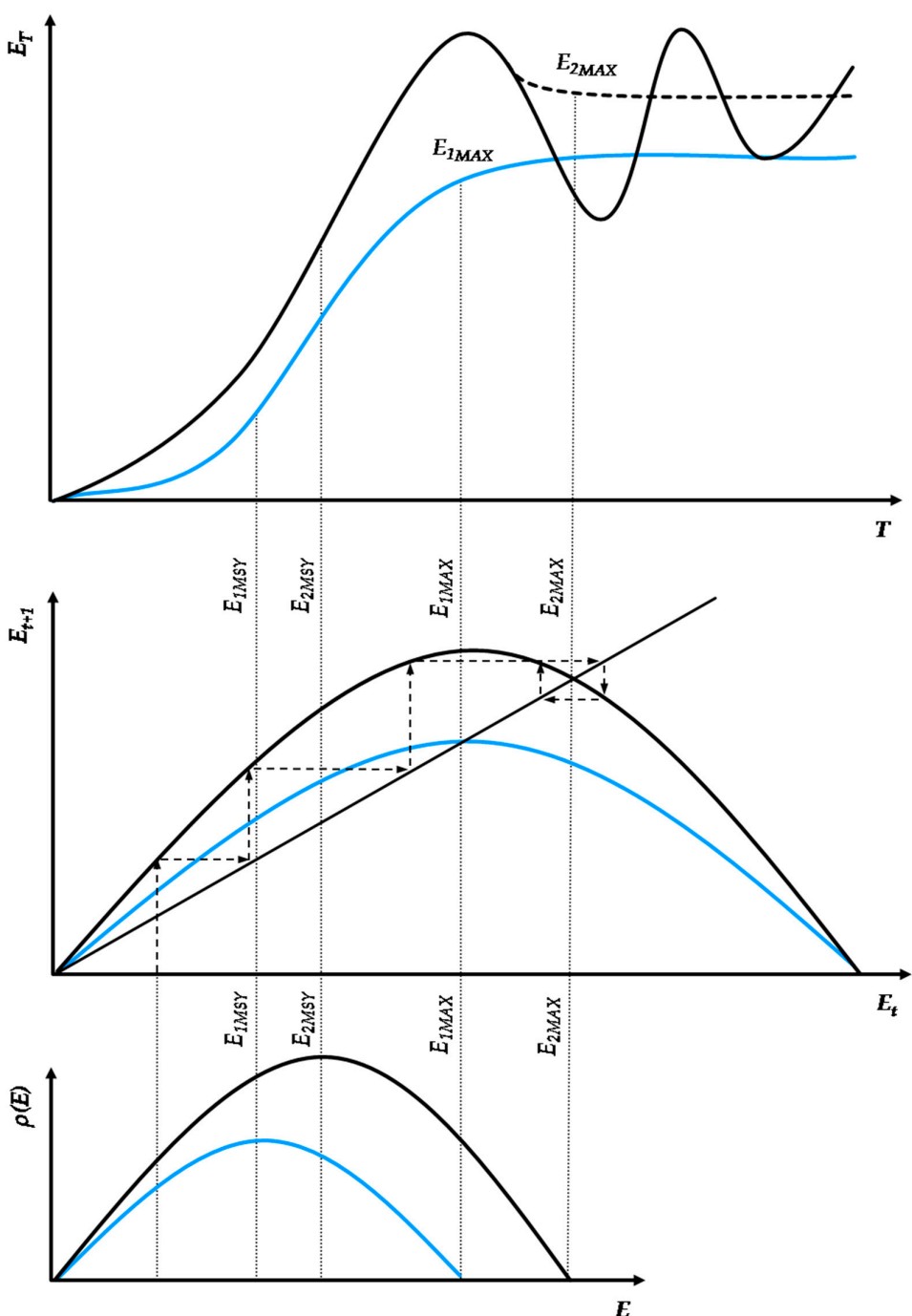

**Figure A1.** Graphical depiction of the *Logistic Cobweb Map* with two growth paths: (a) *Optimally Stable* (blue) and (b) *Convergent Stable* (black).

In Figure A1, the three elements forming the *Logistic Cobweb Map* are presented from a bottom-up view. The logistic cobweb map is fundamental; thus, typically, all of its other mathematical versions are built with the same rationale. Regarding the map's structure, instead of a typical ecosystem population $x_t$ with discrete individuals, we consider the EEP $E_t$ to be as it is defined in Equation (1). The bottom graph represents the growth rate $\rho(E_t)$ at each $E_t$ between zero (=0) and the *maximum stable EEP* [$E_t \in [0, \text{E}_{MAX(S)}]$]. The maximum stable EEP is defined as such due to the fact that the EEP may be unstable; hence, it is likely to meet $E_{MAX(S)}$ by converging to it, fluctuating around it at constant periods, or even with divergence. In any case, at $E_{MAX(S)}$, the growth rate is zero, and the system is stabilized. The middle graph depicts the core cobweb plot with the analytical view of the growth path

between any $E_{t-1}$ and $E_t$, starting from any initial non-zero value of $E_0$. This plot consists of the map of Equation (A1) and the 45° line where a population at time step $t-1$ is inserted into Equation (A1) in line with the rationale of Equation (8), $E_t = f(E_{t-1})$. The upper graph represents the temporal evolution of the $E_T$ size at each time step $T$.

The elements of Figure A1 reflect the mathematical properties of the reproduction function as formulated in Equation (A1). By substituting $x$ with $E$, we may write Equation (A1) again in terms of EEP at each time step $t$ as:

$$E_t = a \cdot E_{t-1} \cdot \left(1 - \frac{b}{a} \cdot E_{t-1}\right), \quad a > b, a, b, E \in R^+, t \in N^+ \tag{A3}$$

In Figure A1, two cases of Equation (A3) are presented. The first and simplest case is depicted by an *optimally stable system* (blue line), meaning that the parameter values $a,b$ of Equation (A3) are such so that the system not only stabilizes smoothly (without fluctuations) but also stabilizes at its *peak stable population size*. Specifically, this means that the 45° line meets the core cobweb plot at its maximum point, which can be found from its first derivative:

$$\frac{df(E)}{dE} = f'(E) = a - 2 \cdot b \cdot E, \quad a > b, a, b, E \in R^+, t \in N^+ \tag{A4}$$

According to Equation (A4), the maximum of the core cobweb plot is found for:

$$E_{MAX} = a/2 \cdot b, \quad a > b, a, b, E \in R^+ \tag{A5}$$

According to Equation (A5), this is the global maximum of the core cobweb plot as the second derivative is always negative for all $E \in R^+$. Specifically:

$$\frac{d^2 f(E)}{dE^2} = f''(E) = -2 \cdot b, \quad b \in R^+ \tag{A6}$$

Equations (A4)–(A6) have both a mathematical and economic influence on the EEP growth path with respect to maximization and stability. As, from an ecological perspective, *a civilization is a social system aiming at the maximization of its energy use to support its growing structural complexity constrained by carrying capacity resource depletion*, an emerging question in the context of the logistic cobweb map modeling is *what its intrinsic growth rate should be to arrive at its maximum theoretical EEP without fluctuations*? This holds significant economic meaning as a social system evolves via generating innovations regarding more sophisticated ways of harvesting and utilizing energy from the environment, and favoring more gradual EEP growth that would, in turn, derive from a lower intrinsic growth rate. By applying this rationale to the above mathematical framework, we further combine Equations (A1) and (A5) by solving for $E_t = a/2b$. Specifically, we may write:

$$a_{\overline{E_{MAX}}} = a \cdot \left(\frac{a}{2b}\right) \cdot \left(1 - \frac{b}{a} \cdot \left(\frac{a}{2b}\right)\right), \quad a > b, a, b \in R^+ \tag{A7}$$

By solving Equation (A7), the universally optimal value of the intrinsic growth rate to stabilize the $E_t$ at the map's maximum point ($E_{MAX}$ capped) without fluctuations for $a = 2$ is as shown in Figure A1 (in blue). Any other value below or above $a = 2$ will have different impacts on stability level of the EEP. Specifically, for any value $a < 2$, the system will stabilize smoothly but at a lower EEP level than the maximum stability. For any value $a > 2$, the system will be *non-invertible with memory loss* [72]. Specifically, it may be convergent-stable (for $2 < a < 3$, as shown in the black line in Figure A1) or fluctuate permanently (for $3 < a < 4$) or become chaotic (for $a > 3.8$) or collapse (for $a \geq 4$). In our work, we will skip scrutinizing the mathematical conditions of the logistic cobweb map's stability, as we are focusing on the facets with economic meaning. However, a complete presentation of such issues can be found in Robert May's classical work [69] on the logistic map as well as of

other authors for a generalized theory of complex systems' stability [73] and *thermodynamic non-invertibility* [72], where the logistic map is also put under such perspective.

From Equation (A7), we may also conclude that the value of parameter *b* has no effect on the system's stabilization at the maximum stable population, as, by solving Equation (A7) for *a* = 2, we find:

$$\overline{E_{MAX}} = 1/b, \quad b \in R^+ \tag{A8}$$

Equation (A7) suggests that while, for *a* = 2, the system will stabilize at its maximum stable population, *the size of this population* will depend exclusively on the value of parameter *b*. Indeed, according to Equation (A2) the ratio *a/b* defines the system's carrying capacity *K*; hence, assuming a constant value of parameter *a* = 2, the carrying capacity that is equivalent to the system's maximum EEP will increase as the value of *b* becomes lower, meaning that *the lower the population growth limitation intensity coefficient is, the higher will the system's population growth potential be*. Again, we may identify an economic meaning in this condition as well: *the more efficient is the system in utilizing its (energy) resources, the more (energy equivalent) individuals it will be able to support*.

A last remark on the system's stability concerns the second case presented in Figure A1, concerning *convergent stability* (black line), which is also presented more thoroughly. Beginning from any initial $E_0$ condition with a specific $\rho(E_0)$, we observe that, for the same value of parameter *b* and a value of parameter 2 < *a* < 3, the growth path fluctuates around the maximum population size, where, after a sequence of decaying fluctuations, it converges to it (dotted line in the temporal population growth plot). What is of importance here is that, while parameter *a* value deviates from the optimal stability state of *a* = 2 towards higher values, setting the conditions for increasingly unstable population growth, the maximum stable EEP where it could potentially stabilize perpetually is *not impossible but highly unlikely*. There exists *at least one growth path sequence* (beginning from an initial population $E_0$) that satisfies the condition $E_0 \rightarrow E_1 \rightarrow E_2 \rightarrow E_3 \rightarrow \ldots \rightarrow E_{MAX(S)} \forall a \in (1, +\infty)$. Alternatively, by setting an optimization target for the growth path to reach $E_{MAX(S)}$, we could find any previous $E_t$ up to the initial $E_0$ that generates this optimal sequence. Otherwise, this suggests the need of EEP control measures in the real world to compensate for the system's excessively high intrinsic growth rate. The main difference with the optimally stable EEP for *a* = 2 is that the carrying capacity for *a* > 2 is formed at a higher level, as Equation (A2) suggests. Hence, another feature of the logistic growth cobweb model is that *there is a trade-off between stability and EEP size for increasing values of a > 2*.

Aside from the stability facets, we are concerned with the minimum value of parameter *a* so that the EEP stabilizes at a non-zero level. With this approach, the core cobweb plot must possess at least one point higher than the 45° line. If such a condition applies, the *Maximum Sustainable Yield* (MSY) can be calculated as the maximum difference:

$$E_{MSY} = \frac{d(f(E) - E)}{dE} = a - 2 \cdot b \cdot E - 1, \quad a > b, a, b, E \in R^+ \tag{A9}$$

According to Equation (A9), the MSY of the EEP is found for:

$$E_{MSY} = \frac{a - 1}{2b}, \quad a \in (1, +\infty), a > b, b, E \in R^+ \tag{A10}$$

From Equation (A10), we can conclude that in order for any EEP to grow above zero level—whether stable and sustainable (1 < *a* ≤ 2), unstable and sustainable (2 < *a* < 4), or unstable and unsustainable (*a* ≥ 4)—the fundamental condition is a value *a* > 1. This does not cancel the previous conditions set for parameter *a* by Equations (A1)–(A7), as even theoretically, there can be observed population systems that are unsustainable due to insufficient intrinsic growth coefficients. In any case, Equation (A10) narrows the range of values of parameter *a* for positive long-term system growth.

*Appendix A.2. The Logistic Growth Map*

The second discrete-time map examined is the *Logistic Growth Map*. Its properties are quite similar to that of the logistic cobweb map. In general, it has the same mathematical structure and graphical depiction as the cobweb map, with main difference being that it is a *purely growth function* as it contains the independent term $E_{t-1}$, adding the cobweb map (as formulated in Section Appendix A.1) as its *differential*. Specifically, the mathematical formulation of the logistic growth map is:

$$E_t = E_{t-1} + a \cdot E_{t-1} \cdot \left(1 - \frac{b}{a} \cdot E_{t-1}\right), \quad a > b, a, b, E \in R^+, t \in N^+ \tag{A11}$$

Similarly to Equation (A5), the maximum point of the core cobweb plot is for:

$$E_{MAX} = \frac{a+1}{2 \cdot b}, \quad a > b, a, b, E \in R^+ \tag{A12}$$

Equation (A12) gives the global maximum of the logistic growth map as the second derivative, which is always negative for all $E \in R^+$. Specifically:

$$\frac{d^2 f(E)}{dE^2} = f''(E) = -2 \cdot b, \quad b \in R^+ \tag{A13}$$

By solving Equations (A11) and (A12) for $E_t = (a + 1)/2b$, the optimal value of parameter $a$ to stabilize the $E_t$ at the map's maximum point ($E_{MAX}$ capped) without fluctuations is $a = 1$. For this value of parameter $a$, the maximum stable population will be for:

$$\overline{E_{MAX}} = 1/b, \quad b \in R^+ \tag{A14}$$

Based on the logistic growth map's cobweb plot, the MSY is calculated as the maximum difference between the map's function and the 45° line for:

$$E_{MSY} = a/2 \cdot b, \quad a > b; a, b, E \in R^+ \tag{A15}$$

The elements of the logistic growth map have the same general shape as the logistic cobweb map, with only different optimization values due to its additional term $E_t$.

*Appendix A.3. The Beverton–Holt Map (Hassel Map (c = 1))*

Another important family of discrete-time models derives from the *Hassel Map* with three parameters [59] with a very flexible model behavior. The higher limiting case of the Hassel model is with parameter $c = 1$, which gives the *Beverton-Holt Map* as:

$$E_t = \frac{a \cdot E_{t-1}}{(1 + b \cdot E_{t-1})^c}, \quad a > b, a, b, E \in R^+, c = 1, t \in N^+ \tag{A16}$$

As in Equations (A5) and (A12), the maximum population size for the Beverton-Holt map is found for:

$$E_{MAX} = \frac{a-1}{b}, \quad a \in (1, +\infty), a > b, b, E \in R^+ \tag{A17}$$

Based on Equation (15), the *Beverton-Holt Map* dynamics are graphically presented in Figure A2.

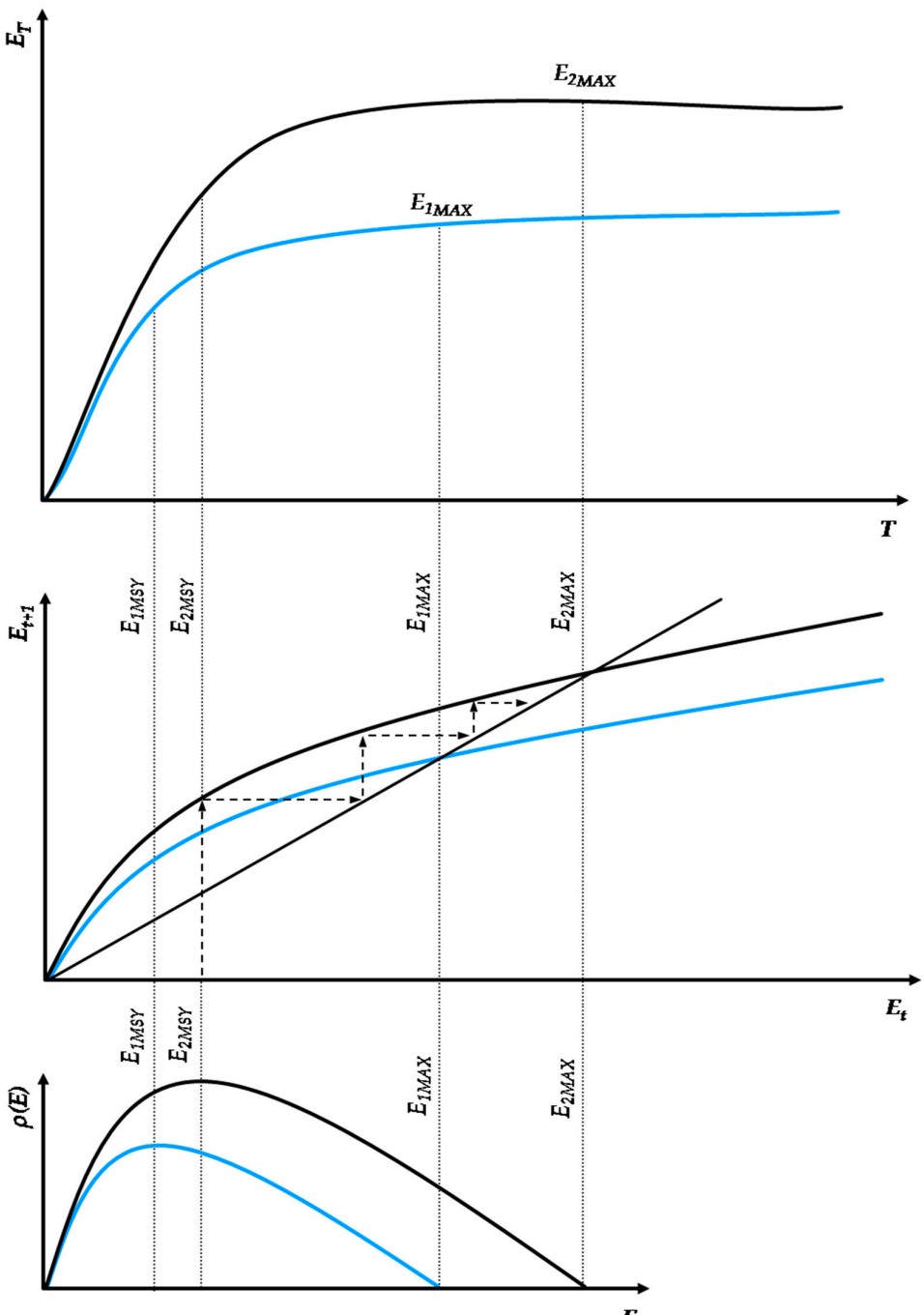

**Figure A2.** Graphical depiction of the constantly monotonic and stable *Beverton–Holt Map* with two growth paths: (a) *low carrying capacity* (blue) and (b) *high carrying capacity* (black).

Equation (A17) gives the Beverton–Holt growth global maximum as the second derivative, which is always negative for all $E \in R^+$. Specifically:

$$\frac{d^2 f(E)}{dE^2} = f''(E) = -b, \quad b \in R^+ \tag{A18}$$

By solving Equation (A16) for $E_t = (a - 1)/b$, we find that the optimal value of parameter $a$ to globally stabilize the $E_t$ is $a = 1$. However, as the Beverton–Holt is a globally monotonic model, value $a = 1$ gives an unsustainable population for any initial value of $E_t$; so based on Equation (A16), the only population size intersecting the 45° line is $E_t = 0$. Hence, any

non-zero sustainable population for the Beverton–Holt model requires the condition of $a > 1$ so that the maximum stable population will be for:

$$\overline{E_{MAX}} \propto (a/b), \quad \forall a \in (1, +\infty), a > b, b, E \in R^+ \tag{A19}$$

Similarly to other maps, for the Beverton–Holt cobweb plot, the MSY is calculated as the maximum difference between the map's function and the 45° line as follows:

$$E_{MSY} = \frac{a-1}{2 \cdot b}, \quad \forall a \in (1, +\infty), a > b, b, E \in R^+ \tag{A20}$$

Based on Equation (A16), the Beverton–Holt map is the only population map differentiating from the rationale presented in Figure 11, describing population growth as a *ratio* instead of *carrying capacity depletion*. It is the only map that is both *globally monotonic and stable* for any combination of parameters *a,b*, with its maximum population size depending exclusively on the value of $a/b$. In simple terms, this is interpreted as the highest possible value of parameter *a* for a fixed parameter *b* value. The Beverton–Holt map version of the Hassel family incorporates many interesting mathematical properties concerning *intra-social contest competition* translated in its globally stable behavior [59,70], as discussed in Section 4.2 of the main text.

*Appendix A.4. The Ricker Map*

The last examined map with two parameters deriving from the Hassel Map as one of its limiting cases is the *Ricker Map* [59,61] as:

$$E_t = a \cdot E_{t-1} \cdot e^{-b \cdot E_{t-1}}, \quad a > b, a, b, E \in R^+, c = 1, t \in N^+ \tag{A21}$$

An important aspect of the Ricker map is that it has been reformulated to generally apply as the *Halter-Transcendental* agricultural production function [63,64] with the same properties to the *Gamma Function*. For a single production factor *X*, the Halter production function is written as a product combination of a *power* and an *exponential* element as:

$$Y = a \cdot X^c \cdot e^{-b \cdot X}, \quad a > b, c > b, a, b, c, X, Y \in R^+ \tag{A22}$$

In Equation (A22), the growth part is expressed by an exponent *c* as a power function, where $c = 1$ gives the *Ricker Map* as special linear intrinsic growth case, similar to the other examined maps. The maximum point of the Ricker map's core cobweb plot is:

$$E_{MAX} = 1/b, \quad b \in R^+ \tag{A23}$$

Equation (A23) suggests that the population at which the map maximizes *depends on the value of parameter b exclusively*, irrespective of the parameter *a* value.

Accordingly, we may identify if the maxima in the Ricker Map are unique by calculating the second derivative as:

$$\frac{d^2 f(E)}{dE^2} = f''(E) = a \cdot b \cdot (b \cdot E - 2) \cdot e^{-b \cdot E}, \quad a > b, b, E \in R^+ \tag{A24}$$

Based on Equation (A24), the second derivative's root is $E_t = 2/b$ at a population size double than the maximum of Equation (A23). Although, from the value $E_t = 2/b$, the map is decreasing at a diminishing rate, the maximum of Equation (A23) is unique as, for any point other than $1/b$, the Ricker map has lower values $\{E(1/b) > E_t, \forall E \in R^+ - [1/b]\}$. The Ricker map is the most complex of the four examined. It is asymmetrical due to the exponential decay part prevailing on the *growth* after value $1/b$ and on the *growth rate* after value $2/b$. As

in all other maps, any non-zero sustainable population requires a value for parameter $a > 1$, meaning that the maximum stable population will be:

$$\overline{E_{MAX}} = (a/b) \cdot e^{-1}, \quad \forall a \in (1, +\infty), a > b, b, E \in R^+ \tag{A25}$$

According to Equation (A21), the *Ricker Map* dynamics are graphically presented in Figure A3.

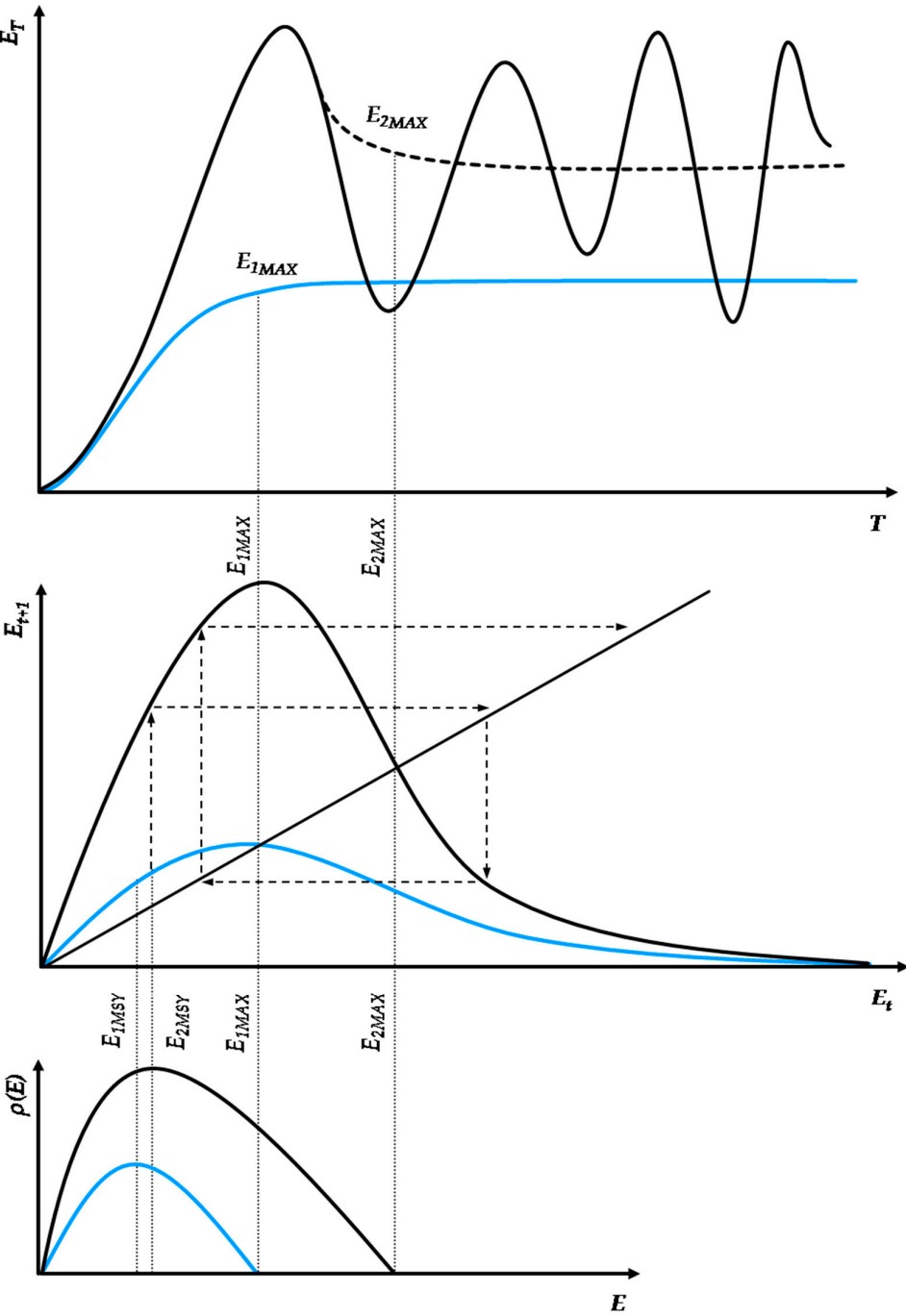

**Figure A3.** Graphical depiction of the *Ricker Map* with two growth paths: (a) *optimally stable* (blue) and (b) *periodic unstable* (black).

From Equation (A25), the optimally stable maximum population is for $a = e \sim 2.71$. In accordance with the other three maps, the MSY for the Ricker cobweb plot is calculated as the maximum difference between the map's function and the 45° line for:

$$E_{MSY} = \frac{W\left(\frac{e}{a}\right) - 1}{b}, \quad a > b, a, b, E \in R^+ \tag{A26}$$

According to Equation (A26), *W(e/a)* is the *Lambert W-Function* [74], which is the basis for numerous transcendental equations and, in its general form, for the real numbers' axis (*R*) is:

$$f(W) = W \cdot e^W, \quad W \in R \tag{A27}$$

The Ricker version of the Hassel family incorporates many interesting mathematical properties concerning *intra-social scramble competition*, which are manifested through its tendency to become unstable for parameter *a* values $a > 2.71$. Excluding the Beverton–Holt model, which is globally stable, the Ricker map is the most resilient in relation to the logistic cobweb and the logistic growth map as its *optimally stable maximum population* can be achieved with a higher value of parameter *a*, although this provides a lower population size due to its exponential decay part. Due to its exponential part though, the Ricker map fundamentally differs from the other two maps in that it is *globally sustainable* for any value of parameter *a* irrespective of its instability intensity $[\forall a \in (1, +\infty)]$. Mathematically, after the $E_t = 2/b$ value the exponential decay behavior prevails in the map, it will *asymptotically converge* to zero population size but practically never be equal to it. Hence, irrespective of how much the population deviates from maximum stability due to an extremely high parameter *a* value, the feedback loops of Equation (A21) will always yield a positive but periodically very low population. As discussed in Section 4.2, this behavior of scramble competition suggests that resources are accessible to *all competitors* without allocation rules by a social hierarchy. With scramble competition, the per capita availability becomes increasingly lower across population growth, as due to free-access, issues of *tragedies of the commons* [75] emerge. It is not surprising that the Ricker map has been extensively used for fish populations that comprise one of the most indicative cases of the tragedy of commons. However, scramble competition additionally suggests the existence of high *resilience* when the population becomes extremely low.

## Appendix B. Energy, Intrinsic Growth Rate and Carrying Capacity

As shown in Appendix A, the value of parameter *a* is crucial for the system's evolution regarding its stabilization above the universal (thermodynamic) equilibrium of zero population. As demonstrated in Section 3.1, setting a constraint on the value of parameter *a* significantly affects the accuracy limits of the tested models in $R^2$ terms. For instance, the 10,000 BC–1800 AC EEP by reconstructed HYDE 3.2 data shows no fluctuations in between, suggesting that $a \leq 2$, as assumed in the "constrained *K*" model version. However, this is a detail that is almost impossible to make known on a global scale due to numerous uncertainties taking place for such a long period, such as the net effect of simultaneous existence of social thrives and collapses in different geographical areas, local diets across the planet's climate zones, the differences in the levels of technology, crop failures, war, pestilence, the shifts of limiting factors, as well as of many other variables. A facet of major importance for the logistic cobweb and the logistic growth maps—as discussed in Section Appendix A.1—is that for populations with unstable but sustainable intrinsic growth rates $2 \leq a < 4$, there exists *at least one possible growth path that leads to a long-term stable population*. However, as the value of parameter *a* keeps increasing, the probability of following that path from a positive population $E_0$ is decreasing. This path exists even for unsustainable populations with $a > 4$, however it is so unlikely that its probability is asymptotically equal to zero. Assuming that, irrespective of its stability, a population is sustainable for values $2 \leq a < 4$, in Figure A4 below, we examine two dimensions of the parameter's value as

it increases for $a \in [2,4]$: its theoretical maximum stable population and its percentage increase in relation to the optimally stable population for $a = 2$.

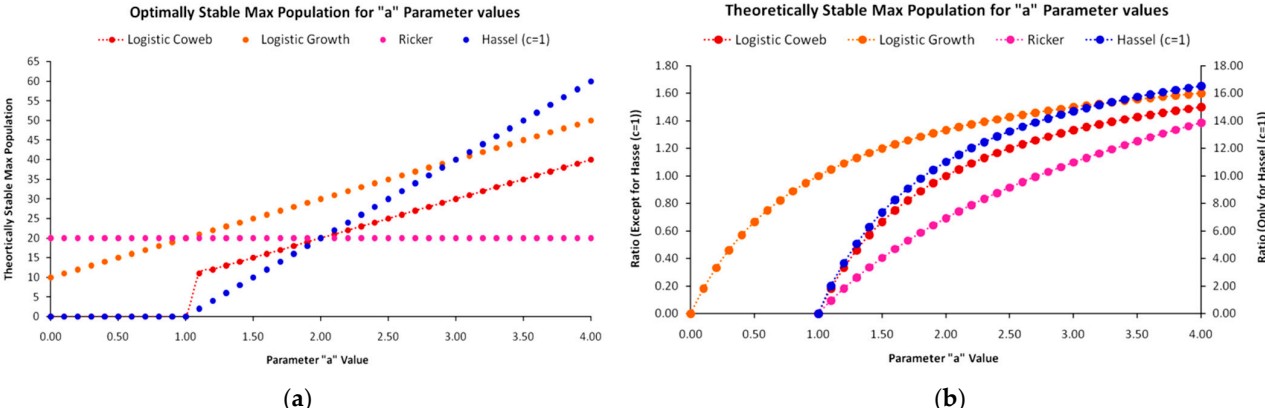

(**a**)                                                                                                 (**b**)

**Figure A4.** The effect of an increase in parameter $a$ value for a constant value of parameter $b = 0.05$ on: (**a**) the increase in the population size at which stabilization *is* achieved (for $a \in (1, 2]$) and theoretical stabilization that *would be* achieved if the optimal growth path was known to lead to this population size (for $a \in (2, 4)$); (**b**) the fraction (%) of this population to the *optimally stable population* for $a = 2$. The secondary vertical axis refers exclusively to the Beverton–Holt (Hassel $c = 1$) model due to differences in scale with the other three maps.

The mathematical formulation for the graphs of Figure A4, calculating the *Optimally Stable Maximum Population* (OSMP) along with the *Theoretically Stable Maximum Population* (TSMP) to the OSMP fraction for each dynamic map, is presented in Table A1.

**Table A1.** OSMP parameter $a$ values and TSM/OSMP mathematical formulas for each map.

| Population Map | OSMP Formula | OSMP Parameter $a$ | TSMP/OSMP Formula |
| --- | --- | --- | --- |
| Logistic Cobweb | $(a - 1)/2b$ | $a = 2$ | $[(1 - a)/(-b)]/(a/2b)$ |
| Logistic Growth | $(a + 1)/2b$ | $a = 1$ | $[(a/b)]/[(a + 1)/2b]$ |
| Beverton–Holt | $(a - 1)/b$ | $a \propto E_t$ | $[(a-1)/b]/[a/(1 + b \cdot E_{t-1})^2]$ |
| Ricker | $1/b$ | $a = e \sim 2.71$ | $\ln(a)$ |

In Figure A4a, except for the Ricker map, for which the OSMP depends exclusively on the value of parameter $b$, the OSMP for the other three maps increases linearly with the increase in the value of parameter $a$. Additionally, the logistic cobweb, Beverton–Holt, and Ricker graphs intersect at the critical value $a = 2$, which, for the logistic cobweb, signifies the OSMP irrespective of the initial population conditions $E_0$. The exception concerns the logistic growth graph, where it achieves the OSMP at its maximum stability, $a = 1$ (as presented in Appendix A), following parallel course to the logistic cobweb graph for values $a > 1$. As shown in Figure A4b, the TSMP/OSMP ratios for all maps converge (they converge asymptotically for $\lim(a) \to +\infty$) as the value of parameter $a$ increases, except for the Beverton–Holt map, which, due to its constantly monotonic growth, is proportional to population size $E_t$, as shown in Table A1. For simplicity, as the Beverton-Holt map is globally monotonic with no single maximum population size—as suggested by Equation (A19)—to extrapolate its graph in Figure A4b, besides the assumed benchmark parameter $b$ value equal to (=0.05), we also set an $E_t$ reference value (=1).

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
