# Peer review of "Energy and the Macrodynamics of Agrarian Societies"

_land, doi:10.3390/land12081603_

Round 1

Reviewer 1 Report

Thank you very much for inviting me to assess the above-mentioned manuscript submitted to Land. From the research topic, this paper attempts to examine “2 Energy and the macrodynamics of agrarian societies” approaches. 

Firstly I suggest authors to rewrite the abstract to make it more constructive. Abstract should have at least one sentence per each: context and background, motivation, hypothesis, methods, results, conclusions. Add some numbers from your findings and a policy sentence.

INTRO:       i) The introduction part of the study needs improvement and story flow and the authors need to give proper contributions to their study. Ä°i) It is better to affard the contributions of the paper in the introduction part. İii) I would like to suggest that authors should update the literature for intro and all text.

 Literature: i) There is a need to do a more rigorous and systematic literature review. See and kindly add the following papers to the references https://doi.org/10.1080/15567249.2016.1263251

https://doi.org/10.1016/j.intfin.2004.03.006

The authors should clearly mention the literature gap.        

 Conclusions are a bit vague and not convincing to the reader.

It would be appropriate to indicate sharp future research directions and limitations of this at the end of the conclusion section just before references.

Author Response

Response Letter to Reviewer #1

Dear MDPI Land journal reviewers, editors and publishing committee,

We deeply thank you for your contribution via your expertise and observations towards the improvement of the scientific accuracy and conveyance of our manuscript’s work significance to the academic community and other audiences who may utilize its value.

The Authors’ Responses (AR) below concern the observations of Reviewer #1 (R#1) as well as the actions adopted for improving our manuscript. Please note that every reference to our manuscript concerns the revised version, submitted along with the current letter.

R#1: Thank you very much for inviting me to assess the above-mentioned manuscript submitted to Land. From the research topic, this paper attempts to examine “Energy and the macrodynamics of agrarian societies” approaches. 

Firstly I suggest authors to rewrite the abstract to make it more constructive. Abstract should have at least one sentence per each: context and background, motivation, hypothesis, methods, results, conclusions. Add some numbers from your findings and a policy sentence.

AR: We have restructured the abstract qualitatively to make it more specific to the reader, better fitting to the sequence recommended by R#1. Regarding the last part (policy), as our work concerns the identification of a pattern in a historical period without direct policy interest, we preferred to include such a part in the Conclusions in relation to the teachings of this period for today’s global and regional policy.

R#1: Intro: i) The introduction part of the study needs improvement and story flow and the authors need to give proper contributions to their study; ii) It is better to afford the contributions of the paper in the introduction part; iii) I would like to suggest that authors should update the literature for intro and all text.

AR: As the introduction is already ~6 pages, including ~ 60% of the literature used, we have restructured the Introduction’s text to follow part i) suggested by R#1. The Introduction gets from very early quite technical with graphs and concepts used across the document; hence, as far as point ii) is concerned, we spread the remaining 40% of our references in the Materials/Methods, Results and Discussion in a targeted way. As far as point iii) is concerned, we have enriched our literature with some additional references.

R#1:  Literature: i) There is a need to do a more rigorous and systematic literature review. See and kindly add the following papers to the references https://doi.org/10.1080/15567249.2016.1263251; https://doi.org/10.1016/j.intfin.2004.03.006.

AR: Our econometric analysis mainly concerns pattern recognition with the use of ecological dynamic models, using existing reconstructed raw data, as well as performing reconstructions of our own based on rational assumptions. These data for the sub-period 10,000BC-1,000AC are available for intervals of 1,000years. For such big time gaps, the application of Error Correction Models, Granger causality and other panel data econometric methods would only have marginal value added to our analysis with questionable reliability. However, as the issue of relating the shift from organic agriculture to industrial agriculture (~1,800AC) with the intensive use of petrochemical fertilizers is relevant to CO2 emissions at large spatio-temporal scales, we have included reference https://doi.org/10.1080/15567249.2016.1263251 in our Conclusions. In this context, we have made a connection between the future availability of data of smaller time intervals and for more livestock types so that such panel econometric methods can be applied more accurately.

R#1: The authors should clearly mention the literature gap.

AR:  Any changes in the Introduction have included a reference in the current literature gap regarding our work, while it is also reminded in the Conclusions along with our future research and publication targets to contribute to its mitigation.

 R#1: Conclusions are a bit vague and not convincing to the reader.

AR: In relation to the above, our conclusions have both been enriched and rephrased to be more concise and specific.

R#1: It would be appropriate to indicate sharp future research directions and limitations of this at the end of the conclusion section just before references.

AR: Our future research and publication directions in relation to current literature gaps are already mentioned across the manuscript (e.g. in page 17, above Figure 12; page 23-24 below Figure 16) and have also concentrated in separate paragraph in the Conclusions.

We, the authors, are available for any further information and clarification.

Sincerely,

Georgios Karakatsanis and Nikos Mamassis

Reviewer 2 Report

A beautiful theoretical conceptualization of Macrdynamics of energy use in agrarian societies and combining aesthetically different time frames with data sets, anthropological events, and the Second law of thermodynamics to correlate the data points. A clear picture of the evolution of society with the increase in energy and also a historical case study of the Roman Empire and its sunset due to the complexity of the system of distribution of energy among different strata of the population.

I suppose this study can be an outlook for future challenges, Humanity will face.

Reviewer 3 Report

The paper titled as 'Energy and the macrodynamics of agrarian societies' is about This paper explores Leslie White's anthropological theory of cultural evolutionism as a theoretical benchmark to study and generalize the macrodynamics of energy use in agrarian societies. The study covers the period from 12,000 BC to 1,800 AC and classifies the evolution of human civilizations into three phases based on their dominant energy paradigm: hunting-gathering, agriculture, and fossil fuels. The focus is on agrarian societies, which heavily relied on agriculture as their energy paradigm for nearly 14,000 years, involving the transformation of solar energy by photosynthetic life, plants, and land. The paper views agrarian societies as ecosystems with abundant, storable energy inputs but limited land transformation and storage capacity. It examines the optimization of land productivity, livestock power utilization, and caloric value harvested from both arable and grazing land. The research includes the mathematical modeling of selected functions and simulates the increase of energy harvesting from land to understand their macrodynamics' reproduction ability and explore related biophysical and social aspects, such as limiting factors, social hierarchy, and competition.

Overall, the paper is well-organized and well-written. The research design is appropriate for the research question, and the methodology is sound. However, there are some minor grammatical imperfections that need attention. For instance, there is inconsistency in the use of English: some words are written in British English, while others are in American English. One example is the word "Labour/Labor," which exists in both forms. The paper should stick to either British English or American English consistently, not use both interchangeably.

Additionally, there are some grammar and punctuation errors that should be addressed before publication. The entire text needs to be carefully reviewed and corrected to ensure its accuracy and clarity. Some examples are:

Line 83: In regards to > in regard to

Line 98 wild life > wildlife

 Line 795 that starts with "I any case,....'

Overall, the paper is well-organized and well-written. The research design is appropriate for the research question, and the methodology is sound. However, there are some minor grammatical imperfections that need attention. For instance, there is inconsistency in the use of English: some words are written in British English, while others are in American English. One example is the word "Labour/Labor," which exists in both forms. The paper should stick to either British English or American English consistently, not use both interchangeably.

Additionally, there are some grammar and punctuation errors that should be addressed before publication. The entire text needs to be carefully reviewed and corrected to ensure its accuracy and clarity. Some examples are:

Line 83: In regards to > in regard to

Line 98 wild life > wildlife

 Line 795 that starts with "I any case,....'

Round 2

Reviewer 1 Report

accept